# The younger flagellum sets the beat for *Chlamydomonas reinhardtii*

Da Wei[1,2]*, Greta Quaranta[3], Marie-Eve Aubin-Tam[1]*, Daniel SW Tam[3]*

[1]Department of Bionanoscience, Delft University of Technology, Delft, Netherlands; [2]Beijing National Laboratory for Condensed Matter Physics, Institute of Physics, Chinese Academy of Sciences, Beijing, China; [3]Laboratory for Aero and Hydrodynamics, Delft University of Technology, Delft, Netherlands

**Abstract** Eukaryotes swim with coordinated flagellar (ciliary) beating and steer by fine-tuning the coordination. The model organism for studying flagellate motility, *Chlamydomonas reinhardtii*, employs synchronous, breaststroke-like flagellar beating to swim, and it modulates the beating amplitudes differentially to steer. This strategy hinges on both inherent flagellar asymmetries (e.g. different response to chemical messengers) and such asymmetries being effectively coordinated in the synchronous beating. In *C. reinhardtii*, the synchrony of beating is known to be supported by a mechanical connection between flagella; however, how flagellar asymmetries persist in the synchrony remains elusive. For example, it has been speculated for decades that one flagellum leads the beating, as its dynamic properties (i.e. frequency, waveform, etc.) appear to be copied by the other one. In this study, we combine experiments, computations, and modeling efforts to elucidate the roles played by each flagellum in synchronous beating. With a non-invasive technique to selectively load each flagellum, we show that the coordinated beating essentially only responds to load exerted on the *cis* flagellum; and that such asymmetry in response derives from a unilateral coupling between the two flagella. Our results highlight a distinct role for each flagellum in coordination and have implication for biflagellates' tactic behaviors.

**\*For correspondence:**
weida@iphy.ac.cn (DW);
m.e.aubin-tam@tudelft.nl (M-EveA-T);
D.S.W.Tam@tudelft.nl (DSWT)

**Competing interest:** The authors declare that no competing interests exist.

## Editor's evaluation

This important manuscript investigates how the two flagella of *C. reinhardtii*, which have inherently different dynamic properties, synchronize their beating. Through careful, non-invasive experiments, the authors help the field to understand the mechanisms responsible for this synchronization, arguing that these mechanisms are due to unilateral coupling between the two flagella. The data are convincing, and the conclusion about unilateral coupling has been strengthened by additional analysis during revision.

## Introduction

Swimming microorganisms ranging from bacteria (*Berg and Brown, 1972*; *Smriga et al., 2016*) to larger flagellates and ciliates (*Hegemann and Berthold, 2009*; *Ueki et al., 2010*; *Stehnach et al., 2021*) must be able to steer to swim towards desirable environments and away from hazardous ones. Such targeted navigation are known as tactic behaviors. For a specific organism, tactic behavior may be underpinned by distinct mechanisms. For example, bacteria modulate the tumbling rate (*Berg and Brown, 1972*) while flagellates and ciliates modulate the waveform (*Brokaw et al., 1974*; *Gong et al., 2020*; *Gadêlha et al., 2020*; *Bennett and Golestanian, 2015*), amplitude (*Rüffer and Nultsch, 1991*; *Ueki and Wakabayashi, 2017*) and frequency (*Naitoh and Kaneko, 1972*; *Ueki et al., 2010*) of their

**eLife digest** Many single-cell organisms use tiny hair-like structures called flagella to move around. To direct this movement, the flagella must work together and beat in a synchronous manner. In some organisms, coordination is achieved by each flagellum reacting to the flow generated by neighbouring flagella. In others, flagella are joined together by fiber connections between their bases, which allow movement to be coordinated through mechanical signals sent between flagella.

One such organism is *Chlamydomonas reinhardtii*, a type of algae frequently used to study flagellar coordination. Its two flagella – named *trans* and *cis* because of their positions relative to the cell's eyespot – propel the cell through water using breaststroke-like movements. To steer, *C. reinhardtii* adjusts the strength of the strokes made by each flagellum. Despite this asymmetry, the flagella must continue to beat in synchrony to move efficiently.

To understand how the cell manages these differences, Wei et al. exposed each flagellum to carefully generated oscillations in water so that each was exposed to different forces and their separate responses could be measured. A combination of experiments, modelling and computer simulations were then used to work out how the two flagella coordinate to steer the cell.

Wei et al. found that only the *cis* flagellum coordinates the beating, with the *trans* flagellum simply copying the motion of the *cis*. A direct consequence of such one-way coupling is that only forces on the *cis* flagellum influence the coordinated beating dynamics of both flagella. These findings shed light on the unique roles of each flagellum in the coordinated movement in *C. reinhardtii* and have implications for how other organisms with mechanically-connected flagella navigate their environments.

flagellar/ciliary beating. However, these active modulations of motility serve the same goal, that is, to generate a spatially asymmetric propulsive force so the cell can steer.

*Chlamydomonas reinhardtii*, the model organism for studies of flagellar motility, achieves tactic navigation by a fine-tuned differential modulation on its two flagella. Studying this organism offers great opportunities to look into how flagella coordinate with each other and how such coordination helps facilitate targeted steering. *C. reinhardtii* has a symmetric cell body and two near-identical flagella inherited from the common ancestors of land plants and animals (*Merchant et al., 2007*). It swims by beating its two flagella synchronously and is capable of photo- and chemotaxis (*Rüffer and Nultsch, 1991*; *Choi et al., 2016*). For this biflagellated organism, effective steering hinges on both flagellar asymmetry and flagellar coordination. On the one hand, the two flagella must be asymmetric to respond differentially to stimuli (*Rüffer and Nultsch, 1990*; *Rüffer and Nultsch, 1991*); on the other hand, the differential responses must be coordinated by the cell such that the beating would remain synchronized to guarantee effective swimming. Understanding this remarkable feat requires knowledge about both flagellar asymmetry and coordination.

The two flagella are known to be asymmetric in several, possibly associated, aspects. First of all, they differ in developmental age (*Holmes and Dutcher, 1989*; *Dutcher and O'Toole, 2016*). The flagellum closer to the eyespot, the *cis*(-eyespot) flagellum, is always younger than the other one, the *trans*(-eyespot) flagellum. This is because the *cis* is organized by a basal body (BB) that develops from a pre-matured one in the mother cell; and this younger BB also organizes the flagellar root (D4 rootlet) that dictates the eyespot formation (*Mittelmeier et al., 2011*). Second, the two flagella have asymmetric protein composition (*Sakakibara et al., 1991*; *Mackinder et al., 2017*; *Yu et al., 2020*). For example, the *trans* flagellum is richer in CAH6, a protein possibly involved in $CO_2$ sensing (*Choi et al., 2016*; *Mackinder et al., 2017*). Finally, the flagella have different dynamic properties (*Kamiya and Witman, 1984*; *Okita et al., 2005*; *Takada and Kamiya, 1997*). When beating alone, the *trans* beats at a frequency 30–40% higher than the *cis* (*Kamiya and Hasegawa, 1987*; *Rüffer and Nultsch, 1987*; *Okita et al., 2005*; *Wan et al., 2014*); the *trans* also displays an attenuated waveform (*Leptos et al., 2013*) and a much stronger noise (*Leptos et al., 2013*; *Wan, 2018*). Additionally, their beating dynamics is modulated differentially by second messengers such as calcium (*Kamiya and Witman, 1984*; *Okita et al., 2005*) and cAMP (*Saegusa and Yoshimura, 2015*).

Remarkably, despite these inherent asymmetries, *C. reinhardtii* cells establish robust synchronization between the flagella. Such coordination enables the cells to swim and steer efficiently, and is mediated by the fibrous connections between flagellar bases (*Quaranta et al., 2015*; *Wan and*

*Goldstein, 2016*). Intriguingly, in the coordinated beating, both flagella display dynamic properties, that is, flagellar waveform, beating frequency (~50 Hz), and frequency fluctuation, that are more similar to those of the *cis* flagellum (*Rüffer and Nultsch, 1985*; *Kamiya and Hasegawa, 1987*; *Leptos et al., 2013*; *Wan et al., 2014*; *Wan, 2018*). This has led to a long-standing hypothesis that 'the *cis* somehow tunes the *trans* flagellum' (*Kamiya and Hasegawa, 1987*). This implies that the symmetric flagellar beating ('breaststroke') observed is the result of interactions between two flagella playing differential roles in coordination. How does the basal coupling make this possible? Recent theoretical efforts show that the basal coupling can give rise to different synchronization modes (*Klindt et al., 2017*; *Liu et al., 2018*; *Guo et al., 2021*); and that flagellar dynamics, such as beating frequency, may simply emerge from the interplay between mechanics of basal coupling and bio-activity (*Guo et al., 2021*). Yet, most theoretical efforts examining flagellar synchronization have assumed two identical flagella, limiting the results' implication for the realistic case. Moreover, very few experiments directly probe the flagella's differential roles during synchronous beating (*Wan and Goldstein, 2014*). Therefore, flagellar coordination in this model organism remains unclear. To clarify the picture experimentally, one needs to selectively force each flagellum, and characterize the dynamics of the flagellar response.

In this study, we address this challenge and devise a non-invasive approach to apply external forces selectively on the *cis*- or the *trans* flagellum. Oscillatory background flows are imposed along an angle with respect to the cell's symmetry axis. Such flows result in controlled hydrodynamic forces, which are markedly different on the two flagella. With experiments, hydrodynamic computations, and modeling, we show definitively that the two flagella are unilaterally coupled, such that the younger flagellum (*cis*) coordinates the beating, whereas the elder one simply copies the dynamic properties of the younger. This also means that only external forces on the *cis* may mechanically fine-tune the coordination. We also study the effect of calcium in the *cis*' leading role as calcium is deeply involved in flagellar asymmetry and hence phototactic steering. In addition, a well-known mutant that lacks flagellar dominance (*ptx1*) (*Horst and Witman, 1993*; *Okita et al., 2005*) is examined. Results show that the coordinating role of *cis* does not need environmental free calcium, whereas it does require the genes lost or mutated in *ptx1*. Our results discern the differential roles of *C. reinhardtii*'s flagella, highlight an advanced function of the inter-flagellar mechanical coupling, and have implications for biflagellates' tactic motility.

## Methodology

We set out to establish a non-invasive experimental technique that exerts differential loads on the flagella of *C. reinhardtii*. Following *Quaranta et al., 2015*, we use external background flows to exert hydrodynamic forcing on captured cells. Hydrodynamic forces are generated by a relative motion between the captured cell and its surrounding fluid. While the cell is captured by a glass pipette fixed in the laboratory frame, the fluid moves with the flow chamber - which is fixed on an oscillating piezo-electric stage. By programming the sinusoidal oscillations of the stage, we generate flows of desired amplitude $U_0$ and frequency $f_0$ and impose these flows along different directions $\theta$ to differentially load the two flagella, see *Figure 1A*. For each recording, we extract the phase dynamics of flagellar beating from videography (*Quaranta et al., 2015*; *Wei et al., 2019*; *Wei et al., 2021*), *Figure 1B*. First, recordings are masked and thresholded to highlight the flagella. Then, we extract the mean pixel values over time within two sampling windows (*Figure 1C*), and convert the two signals into observable-invariant flagellar phases $\phi_{c,t}$ (*Kralemann et al., 2008*), see *Figure 1D*. In all our experimental recordings, the *cis* and the *trans* flagella beat synchronously. Typical phase locking between them is represented in *Figure 1D* inset, where the phase difference $\Delta = \phi_c - \phi_t$ fluctuates around 0. Therefore, $\phi_c$ and $\phi_t$ are equal and denoted as $\phi$.

The response of the flagella to the external hydrodynamic loads is therefore characterized by the difference between the flagellar phase $\phi(t)$ and the phase of the oscillating flow $\phi_0 = 2\pi f_0 t$. Hereafter, we refer to the synchronization of the flagellar beating with the imposed external flow as 'flow entrainment', in order to avoid confusion with the 'synchronization' between the *cis* and the *trans* flagella. In our experiments, the *cis* and *trans* flagella always beat synchronously, therefore flow entrainment always takes place for both flagella simultaneously, regardless of the flow's direction.

Differential external loads are selectively exerted on the *cis*- and the *trans*-flagellum by imposing oscillatory background flows at an angle with respect to the cell's symmetry axis. We use numerical simulations to quantify the differential loads exerted by flows along different angles θ = 45° , 90°, and

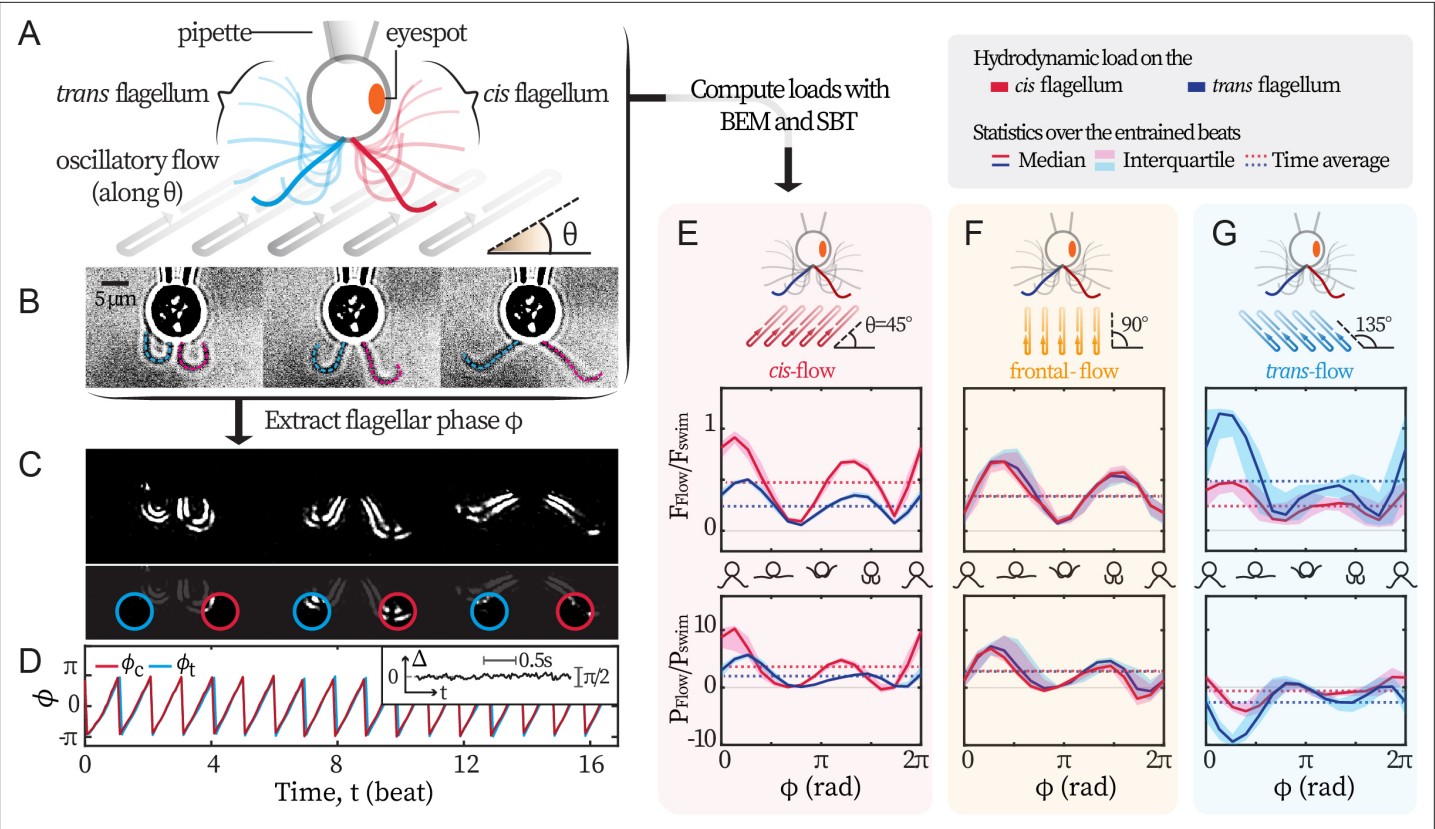

**Figure 1.** Experimental scheme. (**A**) Schematic representation of the experiment. *C. reinhardtii* cells captured by micropipette are subjected to sinusoidal background flows along given directions ($\theta$) in the $xy$-plane. Video recording of flagellar beating (**B**) are thresholded and contrast-adjusted to highlight the flagella (**C**), and mean pixel values within the user-defined interrogation windows (red and blue circles) are used to compute observable-independent flagellar phases (**D**). Inset: flagellar phase difference $\Delta = \phi_c - \phi_t$ of a representative *wt* cell during synchronous beating. Hydrodynamic computations are performed to evaluate the loads due to the background flow on each flagellum. These computations require the flagella to be tracked for the entire duration of the recordings (dashed lines in B) while external flows along a given $\theta$ are applied. (**E, F, G**) represent the hydrodynamic loads for *cis*-, frontal- and *trans*-flows along the $\theta = 45°$, 90° and 135° respectively. Computed forces and the forces' rates of work are presented for the *cis* (red) and the *trans* (blue) flagellum. N≈30 periods during entrainment are used to compute the median (lines) and the interquartile (shadings). Dotted horizontal lines: loads averaged over an entrained beat. Force magnitudes and powers are scaled by $F_{swim}$=9.9 pN and $P_{swim}$=1.1 fW respectively. Flagellar phase corresponds to the displayed shapes in the middle x-axis.

135°, on the *cis*- and the *trans* flagellum, see *Figure 1A*. For each experiment, we track both flagella for the entire duration of the recording, see *Figure 1B*. We use the tracked motion of the flagella together with the oscillatory background flow as the boundary conditions to solve the Stokes equations numerically. Our numerical approach uses boundary element methods (BEM) and slender-body theory (SBT), from which we can directly deduce the total drag force $F$ on each flagellum as well as the associated power $P$ of the viscous forces on each flagellum. In the linear Stokes regime, the drag force $F$ is the sum of the contributions due to the motion of the flagella and the motion of external flow. Our simulations allow us to compute both contributions separately. We present the loads induced by the external flow $F_{Flow}$ and $P_{Flow}$ (see Materials and methods for details). Our computations provide the loads on the flagella throughout the entire experiment, from the time the captured cell is gradually entrained by the external flow, see *Appendix 1: Hydrodynamic computation for asymmetric loading* for the entire time series. *Figure 1E,F,G* represent the loads on each flagellum for flows with $\theta = 45°$, 90°, 135°, after the cell is fully entrained. The variations of the loads are presented for one period of the power-recovery stroke, and have been obtained by phase averaging over N≈30 periods. Upper panels display the magnitude of the drag force $F_{Flow} = |\mathbf{F}_{Flow}|$; while lower panels show associated power $P_{Flow}$. Each panel presents the forces and powers on the *cis* and the *trans*-flagellum in red and blue respectively. The force magnitudes are scaled by $F_{swim} = 6\pi\mu R U_{swim} = 9.9$ pN; while the powers

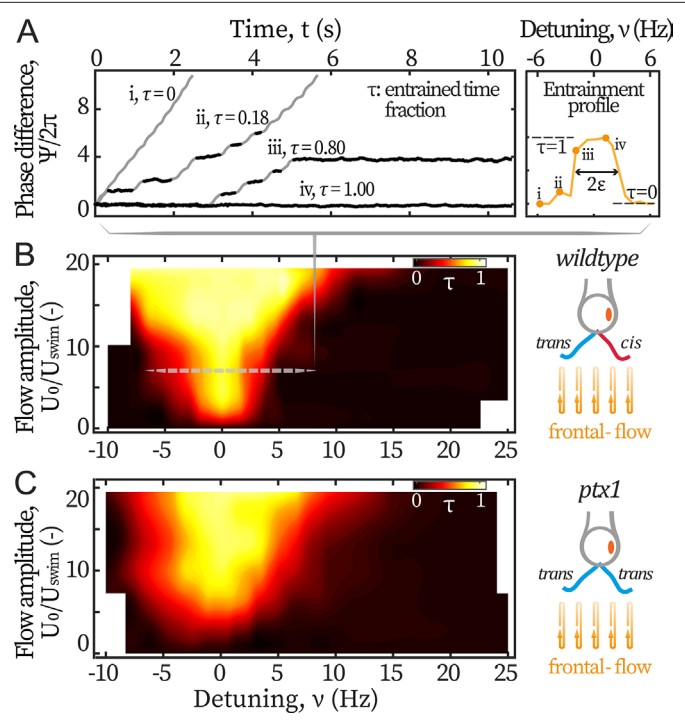

**Figure 2.** Entrainment of *wt* and *ptx1* cells by frontal-flows. (**A**) Phase dynamics of $\Psi = \phi - \phi_0$ at varying detunings $\nu = f_0 - f$. Traces i to iv are taken at detunings marked as shown in the right panel. $\phi$ denotes $\phi_{c,t}$ interchangeably. Plateaus marked black represent flow entrainment ($\dot{\Psi} = 0$) whose time fractions $\tau = t_{entrain}/t_{tot}$ are noted. $t_{tot}$ is the total time of recording. Right panel: The 'flow entrainment profile', $\tau(\nu)$, whose width ($\tau \geq 0.5$) measures the flow's effective forcing strength $2\varepsilon$. Arnold tongue diagram of a representative *wt* cell (**B**) and a representative *ptx1* cell (**C**) tested with frontal-flows. The dashed slice in B corresponds to the data shown in A. The contour is interpolated from N=132 measurements (6 equidistant amplitudes × 22 equidistant frequencies). Color bar for *ptx1*: $\tau = t_{entrain}/t_{IP}$ with $t_{IP}$ the total time of in-phase synchronous beating.

by $P_{swim} = F_{swim} U_{swim} = 1.1$ fW. $F_{swim}$ is the Stokes drag on a typical free-swimming cell (radius $R = 5$ µm, speed $U_{swim} = 110$ µm/s, water viscosity $\mu = 0.95$ mPa·s).

Along θ = 90°, the external flow loads both flagella symmetrically (*Figure 1F*). However, for flows with θ = 45°, the loads on the *cis*-flagellum (red) are ~2 times stronger than those on the *trans* (blue) (*Figure 1E*, $F_{Flow}^c \approx 2F_{Flow}^t$), whereas flows with θ = 135° do the opposite (*Figure 1G*). The selectivity also manifests in (the absolute values of) $P_{Flow}$. We do notice that flows along θ = 135° are able to entrain the flagella with $P_{Flow} < 0$, meaning that the flagella are working against the flows, and this shall be discussed in later sections. Our computations demonstrate that flows along θ = 45° impose stronger loads on the *cis* flagellum, and we will refer to these as *cis*-flows, hereon forward. Likewise, flows on θ = 135° selectively load the *trans* and we denote these as *trans*-flows. Finally, the flows along θ = 90° that approach the cell from the front will be called frontal-flows.

## Results

### Experimental results

#### Frontal-flow entrains both the *wt* and *ptx1* cells

We first study the cells' response to symmetric hydrodynamic loads, i.e., frontal-flows. Captured cells are subjected to flows of various amplitudes $U_0$ varying between 390 and 2340 µm/s, and frequencies $f_0$ varying between 40 and 75 Hz. The scanned range covers reported intrinsic frequencies of both the *cis* and *trans* flagellum (*Kamiya and Witman, 1984*; *Kamiya and Hasegawa, 1987*; *Rüffer and Nultsch, 1987*; *Takada and Kamiya, 1997*), while the amplitude reaches the maximum instantaneous speed of a beating flagellum (~ 2000 µm/s). *Figure 2A* displays representative variations of the phase difference $\Psi(t) = \phi(t) - \phi_0(t)$ between the flagella and the flow, for different frequencies $f_0$

of the forcing (expressed in terms of the detuning $\nu = f_0 - f$, *Pikovsky et al., 2001*). Entrainment of the flagella by the flow is characterized by plateaus in the phase difference $\Psi(t)$, marked in black on *Figure 2A*. This typical dynamics of the phase difference $\Psi(t)$ between the flagella and the external periodic forcing is well captured by the Adler equation (*Pikovsky et al., 2001*; *Polin et al., 2009*; *Friedrich, 2016*):

$$\dot{\Psi} = -2\pi\nu - 2\pi\varepsilon \sin\Psi + \zeta, \qquad (1)$$

with $\varepsilon$ the flow's effective forcing strength. $\varepsilon$ describes the sensitivity of the flagellar phase to the external stimuli, and it thus depends on both the absolute strength of the flow as well as the biological susceptibility of the flagella. In this study, $\varepsilon$ is experimentally measured from the phase dynamics to quantify the flow entrainment effectiveness. $\zeta$ represents a white noise that satisfies $\langle \zeta(t'+t)\zeta(t') \rangle = 2T\delta(t)$, with $T$ an effective temperature and $\delta(t)$ the Dirac delta function.

By definition, entrainment corresponds to a constant phase difference $\Psi$ ($\dot{\Psi} = 0$). Solving *Equation 1* under this condition, one sees that entrainment is only possible when the effective forcing strength is strong enough: $|\varepsilon| > |\nu|$. Therefore, we can experimentally vary $\nu$ and measure $\varepsilon$ directly by the frequency range where flow entrainment is established. The quality of entrainment is described by the entrained time fraction $\tau = t_{\text{entrain}}/t_{\text{tot}}$, where $t_{\text{entrain}}$ is the total time where the beating is entrained and $t_{\text{tot}}$ the flow's duration, see Materials and methods.

In *Figure 2A*, the traces range from: no entrainment ($\tau=0$, *i*) and unstable entrainment ($0 < \tau < 1$, *ii-iii*), to stable entrainment ($\tau=1$, *iv*). In this study, the frequency range of $\nu$ for which $\tau \geq 0.5$ is considered as the region where entrainment is established, and is used as a measure for the flagellum-flow coupling strength $\varepsilon$ (see *Figure 2A* right panel). This method measures $\varepsilon$ accurately when noises are low ($T \lesssim 2.5$ rad$^2$/s for typical values of frequencies and couplings used in this work). In this regime, this straightforward method is equivalent to previous methods based on multi-parameter curve fitting (*Wan et al., 2014*; *Quaranta et al., 2015*) but is more robust (*Appendix 1: Extracting coupling strength by fitting phase dynamics*).

The flow entrainment landscape over the entire scanned ranges is presented in *Figure 2B*. Up until the strongest flow amplitude, the external forces cannot disrupt the synchronized flagellar beating. In addition, entrainment is never established around frequencies other than $f \approx 50$ Hz. Both phenomena indicate that the inter-flagellar coupling is much stronger than the flows' maximum effective forcing ($\varepsilon \sim 10$ Hz, see *Appendix 1: Monte-Carlo simulations*).

We also examine the flagellar dominance mutant *ptx1* whose two flagella are both putatively considered *trans* flagella (*Horst and Witman, 1993*; *Rüffer and Nultsch, 1997*; *Rüffer and Nultsch, 1998*; *Okita et al., 2005*; *Leptos et al., 2013*): they respond similarly to changes of calcium concentrations (*Horst and Witman, 1993*) and have similar beating frequencies when demembranated and reactivated (*Okita et al., 2005*). *Ptx1* mutants have two modes of synchronous beating, namely, the in-phase (IP) mode and the anti-phase (AP) mode (*Rüffer and Nultsch, 1998*; *Leptos et al., 2013*). By applying frontal-flows over the same ranges of frequencies and amplitudes, we find that the IP beating, which is in a similar breaststroke-like pattern to that of *wt* and is also around $f \approx 50$ Hz, is the only mode that can be flow-entrained. We focus on this mode and report $\tau$ as $\tau = t_{\text{entrain}}/t_{\text{IP}}$ for this mutant, where $t_{\text{IP}}$ is the total time of IP-beating under the applied flows, see *Figure 2C*. Albeit noisier, the Arnold tongue of *ptx1* covers a slightly larger width compared to *wt*'s, meaning that the breaststroke beating of two *trans* flagella is similarly entrainable as that by one *cis* and one *trans* (*Figure 2B–C* right panels). This finding indicates that the *trans* flagellum is at least as susceptible to hydrodynamic loads as the *cis*.

### *Cis*-flow entrains *wt* cells more effectively than *trans*-flow

Next, we study how a cell is entrained by asymmetric flagellar loads. To each captured cell, we apply *cis*-flows and *trans*-flows of a fixed amplitude ($\sim 7U_{\text{swim}}$) but at varying detunings (see Materials and methods), and we compare the flows' effective forcing $\varepsilon$ to quantify the cells' differential response.

We find that *cis*-flows are the most effective in entraining the beating (*Figure 3A*). We illustrate this point with the entrainment profiles of an exemplary cell (*Figure 3A* inset). First, although both the *cis*-flow (red) and the *trans*-flow (blue) can entrain the cell at small detunings ($|\nu| < 0.5$ Hz), the *cis*-flow entrainment is more robust and lasts over the entire duration of the experiment ($\tau(cis\text{-flow})=1$), while the *trans*-flow for a smaller time fraction ($\tau(trans\text{-flow}) \approx 0.85$). This is due to phase-slips (step-like

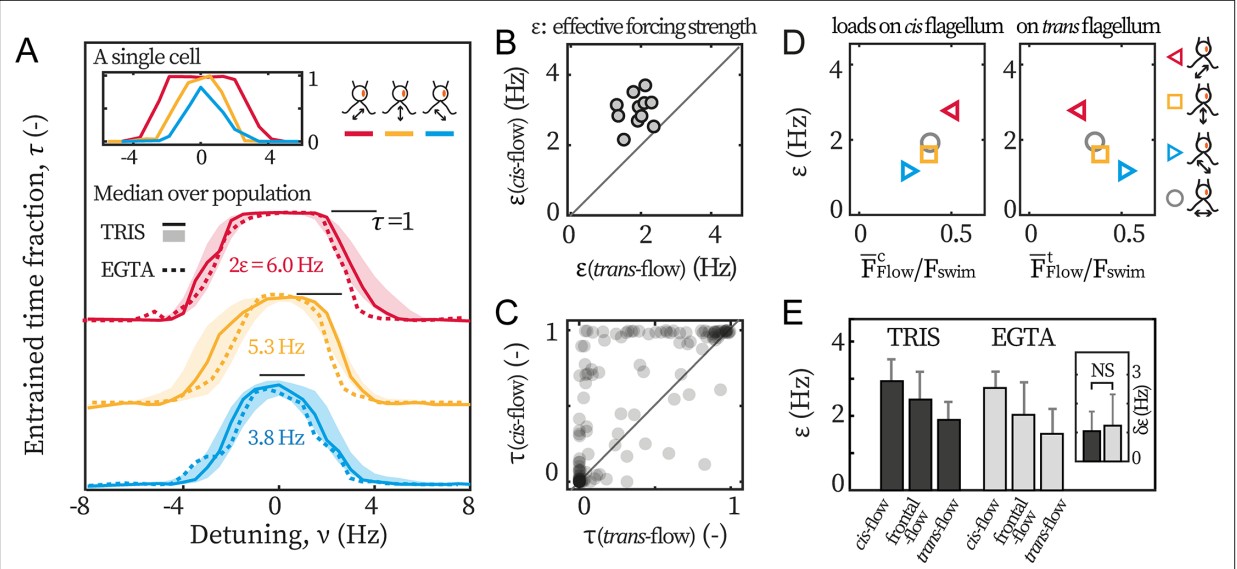

**Figure 3.** *Cis*-flow entrains *wt* cells the most effectively. (**A**) The entrainment profiles $\tau$ of a representative *wt* cell (inset), the median profile of the TRIS group *wt* cells (N=11, solid lines) and the EGTA group (N=6, dashed lines), with either *cis*-flows (red), frontal-flows (yellow) or *trans*-flows (blue). Shaded areas are the interquartile ranges for the TRIS group. (**B**) Comparing $\varepsilon$(*cis*-flow) against $\varepsilon$(*trans*-flow) for each *wt* cells (TRIS group) tested. Solid line: the first bisector line ($y = x$). (**C**) Comparing $\tau$(*cis*-flow) against $\tau$(*trans*-flow) for each cell at each flow frequency. Each point represents the time fractions of a particular cell entrained respectively by *cis*-flow and *trans*-flow of the same frequency. More than 90% of N=132 points are above the first bisector line (solid line). (**D**) Relations between $\varepsilon$ and the loads on the *cis* and the *trans* flagellum. Markers represent different flow directions, see the drawings. (**E**) The effective forcing strengths $\varepsilon$ of the TRIS group (black) and the EGTA group (gray). Bars and error bars: mean and 1 std., respectively. Inset: $\delta\varepsilon = \varepsilon$(*cis*-flow) $- \varepsilon$(*trans*-flow). NS: not significant, p>0.05, Kruskal-Wallis test, one-way ANOVA.

changes in $\Psi$ in *Figure 2A*) between flagella and the flow, and means that the *trans*-flow entrainment is less stable. Additionally, for intermediate detuning (0.5 Hz < |$\nu$| < 4 Hz), $\varepsilon$(*cis*-flow) = $\varepsilon$(*trans*-flow) is always larger than $\tau$(*trans*-flow). In some cases, the *cis*-flow entrains the cell fully whereas the *trans*-flow fails completely (e.g. at $\nu = -2$ Hz). Together, these results imply that a flow of given amplitude entrains flagellar beating more effectively if it selectively loads the *cis* flagellum.

We repeat the experiments with cells from multiple cultures, captured on different pipettes, and with different eyespot orientations (~50% heading rightward in the imaging plane) to rule out possible influence from the setup. The entrainment profile of N=11 *wt* cells tested in the TRIS-minimal medium (pH = 7.0) are displayed in *Figure 3A* (labeled as 'TRIS'). On average, $\varepsilon$(*cis*-flow) = 2.9 Hz and is 70% larger than $\varepsilon$(*trans*-flow) = 1.7 Hz. It bears emphasis that for every single cell tested (11/11), the relation $\varepsilon$(*cis*-flow) > $\varepsilon$(*trans*-flow) holds true. In *Figure 3B*, we show this by representing each cell as a point whose *x*- and *y* coordinates are respectively its $\varepsilon$(*trans*-flow) and $\varepsilon$(*cis*-flow). A point being above the first bisector line ($y = x$) indicates that $\varepsilon$(*cis*-flow) > $\varepsilon$(*trans*-flow) for this cell. All cells cluster clearly above the line. The entrainment asymmetry is very robust and is observed in almost all separate experiments for each cell and each flow condition. In *Figure 3C*, each point corresponds to the time fractions of the same cell entrained by the *cis*-flow and the *trans*-flow at the same frequency. Most points (>90%) are above the first bisector line, meaning that $\tau$(*cis*-flow) > $\tau$(*trans*-flow). Altogether, all results show that selectively loading the *cis* flagellum better enables flow entrainment, pointing to *cis* and *trans* flagella playing differential roles in the synchronous beating.

To highlight the differential roles, we resolve how the flows' effective forcing strengths depend on the actual hydrodynamic loads on each flagellum, see *Figure 3D*. The loads are characterized by the beat-averaged force, $\overline{F}_{\mathrm{Flow}} = \int_0^{2\pi} F_{\mathrm{Flow}} d\phi/2\pi$, on each flagellum (see corresponding horizontal lines in *Figure 1E–G*). These loads are computed for the *cis*-flow, *trans*-flow, and frontal-flow. Experimental and computational data of flows along θ = 0° (circles, *Appendix 1: Hydrodynamic computation for asymmetric loading*) are also included to substantiate the results. From *Figure 3D*, we see that the effective forcing strength scales with the beat-averaged drag on the *cis*, $\varepsilon \sim \overline{F}_{\mathrm{Flow}}^c$ while we find no such correlation between $\varepsilon$ and $\overline{F}_{\mathrm{Flow}}^t$. Notably, the linear relation between $\varepsilon$ and $\overline{F}_{\mathrm{Flow}}^c$ has an intercept near zero ($\varepsilon|_{\overline{F}_{\mathrm{Flow}}^c=0} \approx 0$). Given the total forces on both flagella ($\overline{F}_{\mathrm{Flow}}^c + \overline{F}_{\mathrm{Flow}}^t$) for these flows

remains almost constant (0.74–0.79 $F_{swim}$), the zero-intercept implies that for a hypothetical flow that exerts no load on the *cis* but solely forces the *trans*, it will not entrain the cell at all. This suggests that the hydrodynamic loads on the *trans* flagellum do not significantly contribute to flow entrainment for *wt* cells. We will later discuss the implications of this result in the context of our earlier observation that the *trans* flagella of *ptx1* are susceptible to entrainment by external hydrodynamic loads.

## Depletion of environmental calcium does not affect the asymmetric flow response

We examine whether this newly observed *cis-trans* asymmetry is affected by calcium depletion. Calcium is a critical second messenger for modulating flagellates' motility and is deeply involved in phototaxis (**Yoshimura, 2011**). The depletion of the free environmental calcium is known to degrade flagellar synchronization and exacerbate flagellar asymmetry (**Kamiya and Witman, 1984**). Here, we focus on whether calcium depletion affects the asymmetry $\varepsilon(cis\text{-flow}) > \varepsilon(trans\text{-flow})$. We deplete environmental calcium by EGTA-chelation, following the protocol in **Wakabayashi et al., 2009**. Similar to previous reports (**Kamiya and Witman, 1984**; **Pazour et al., 2005**), the number of freely swimming cells drops significantly in EGTA-containing medium. However, the remaining cells beat synchronously for hours after capture. For these beating cells, calcium depletion is first confirmed by characterizing their deflagellation behavior. Indeed, calcium depletion is reported to inhibit deflagellation (**Wan et al., 2014**; **Quarmby and Hartzell, 1994**). In experiments with standard calcium concentration, all cells deflagellated under pipette suction (20/20). For experiments conducted in calcium depleting EGTA-containing medium, we observe deflagellation to occur in none but one cell (1/19).

After confirming the calcium depletion in our experiments, we perform the same sets of flow entrainment experiments. The dashed lines in **Figure 3A** show the median entrainment profiles for N=6 cells (labeled as 'EGTA'). Clearly, the asymmetry $\varepsilon(cis\text{-flow}) > \varepsilon(trans\text{-flow})$ is unaffected and it again applies for every single cell tested. The mere effect of calcium depletion appears to be a drop in the mean values of $\varepsilon$ (**Figure 3E**). However, the difference between $\varepsilon(cis\text{-flow}) = \varepsilon(trans\text{-flow})$ and $\varepsilon(trans\text{-flow})$ is not affected, see $\delta\varepsilon = \varepsilon(cis\text{-flow}) - \varepsilon(trans\text{-flow})$ for the two experimental conditions in **Figure 3E** inset.

## *ptx1* mutant loses the asymmetric flow response

Lastly, we examine the asymmetry in *ptx1*. The entrainment profiles of *ptx1* are shown in **Figure 4A**. The median profiles are of similar width and height, indistinguishable from each other, and hence

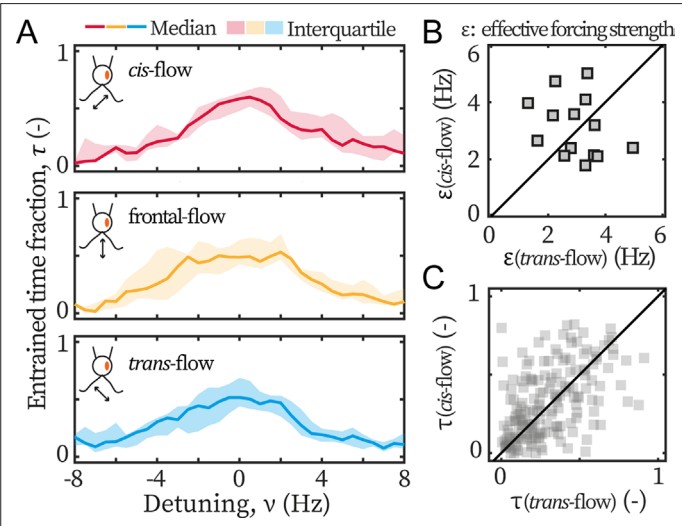

**Figure 4.** *Ptx1* mutant loses the asymmetric flow response. (**A**) Flow entrainment profiles of N=14 *ptx1* cells, tested with *cis*-flows (red), frontal-flows (yellow) and *trans*-flows (blue). (**B**) $\varepsilon(cis\text{-flow})$ and $\varepsilon(trans\text{-flow})$ of the tested cells. The first bisector line (solid): $y = x$. (**C**) $\tau(cis\text{-flow})$ and $\tau(trans\text{-flow})$ for each cell at each applied frequency (N=154 points). $\varepsilon$ and $\tau$ denote respectively the effective coupling strength and the entrained time fraction during IP beating.

indicate a loss of asymmetric susceptibility to flow entrainment, which are in sharp contrast to the profiles of *wt* (*Figure 3A*). The loss is further confirmed by the extracted $\varepsilon$ and $\tau$ presented in *Figure 4B-C*, which are also significantly different from our results for *wt* cells (*Figure 3B-C*). For *ptx1*, cells and entrainment attempts are distributed evenly across the first bisector lines: 7/14 cells are above $\varepsilon(cis\text{-flow}) = \varepsilon(trans\text{-flow})$ in *Figure 4B*, and ~50% points are above $\tau(cis\text{-flow}) = \tau(trans\text{-flow})$ in *Figure 4C*. Altogether, all results consistently show that the asymmetry is lost in *ptx1*.

In addition, it is noteworthy that the measured $\varepsilon$ for both *wt* and *ptx1* are within the same range (2–4 Hz), indicating that both strains are similarly susceptible to external flow. Furthermore, the transition to flow entrainment is sharper for *wt* cells, in *Figure 3A*, compared to *ptx1* cells, in *Figure 4A*. For *wt*, the flow entrainment is robustly established for the entire length of the experiment and $\tau \approx 1$ for a range of $f_0$. For frequencies further away from the intrinsic flagellar beating frequency ($f$), the entrainment time sharply decreases to $\tau = 0$ (*Figure 3A*). For *ptx1* on the other hand, the transition is not as sharp (*Figure 4A*). This difference can be explained by a larger stochasticity in the beating of *ptx1* compared to *wt*, which can be represented by a stronger noise $\zeta$ for *ptx1* compared to *wt* in *Equation 1*.

## Model
### Model experimental findings by three coupled oscillators
To investigate the implications of our experimental results on the coupling between flagella and their dynamics, we develop a model for the system (*Appendix 1: Monte-Carlo simulations*), representing flagella and external flows as oscillators with directional couplings:

$$
\begin{cases}
\dot{\phi}_0 = 2\pi f_0 \\
\dot{\phi}_c = 2\pi[f_c - \lambda_{tc}\sin(\phi_c - \phi_t) - \epsilon_c\sin(\phi_c - \phi_0)] + \zeta_c \\
\dot{\phi}_t = 2\pi[f_t - \lambda_{ct}\sin(\phi_t - \phi_c) - \epsilon_t\sin(\phi_t - \phi_0)] + \zeta_t.
\end{cases}
\tag{2}
$$

$\phi_{0,c,t}(t)$ respectively represent the phase of the flow, the *cis*, and the *trans* flagellum. $f_{0,c,t}$ represents the (inherent) frequency of the flow, the *cis*, and the *trans*, respectively. The first equation represents the imposed periodic forcing, while the other equations represent the Langevin dynamics for the phases of the *cis* and *trans* flagella. The phase dynamics of each flagellum is influenced by the interactions with the other flagellum as well as the interactions with the periodic hydrodynamic forcing. Here, $\lambda_{tc}$ represents the forcing exerted by the *trans* on the *cis* and $\lambda_{ct}$, the forcing of the *cis* on the *trans*. $\epsilon_c$ is the hydrodynamic forcing on the *cis* and $\epsilon_t$ the one on the *trans*. In this simple model, we differentially vary $\epsilon_c$ and $\epsilon_t$ to match the values of the selective hydrodynamic loads ($\overline{F}_{\text{Flow}}^c / \overline{F}_{\text{Flow}}^t$) measured for each flow condition (*Figure 1E–G*). It bears emphasis that $\epsilon_{c,t}$ are input parameters of our model, whereas $\varepsilon$ is measured from the phase dynamics to characterize the entrainment. In our simulations, $\varepsilon$ is extracted following the same approach as in the experiments, see *Figure 2A*. Lastly, $\zeta_{c,t}$ represent the uncorrelated white noises in the *cis* and *trans* flagellum respectively, whose strengths are $\langle \zeta_{c,t}(t' + t)\zeta_{c,t}(t') \rangle = 2T_{c,t}\delta(t)$. *Figure 5A* illustrates our model for flagellar beating subjected to *cis*-flows. The direction and thickness of arrows represent the coupling direction and strength respectively.

*Equation 2* can be combined and readily reduced to a single equation, which has the form of the Adler equation (*Equation 1*) in the asymptotic limit of $\phi_c \approx \phi_t$, see *Appendix 1: Monte-Carlo simulations* for detail. From this equation, we can directly write the quantities $f$, $\varepsilon$ and $T$ measured in our experiments as a function of the parameters from our model $f_{c,t}$, $\epsilon_{c,t}$, $\lambda_{ct,tc}$, and $\zeta_{c,t}$ as

$$
\begin{cases}
f &= \alpha f_c + (1 - \alpha)f_t, \\
T &= \alpha^2 T_c + (1 - \alpha)^2 T_t, \\
\varepsilon &= \alpha \epsilon_c + (1 - \alpha)\epsilon_t,
\end{cases}
\tag{3}
$$

with $\alpha = \lambda_{ct} / (\lambda_{ct} + \lambda_{tc})$. Remarkably, our experimental results are reproduced by the simple theoretical limit of $\alpha \approx 1$, for which the beating dynamics of the cell, frequency $f$ and noise $T$ become equal to those of the *cis* flagellum, and the flow entrainment strength $\varepsilon$ only depends on the forcing strength on the *cis* $\epsilon_c$. $\alpha$ approaching unity corresponds to the limit when $\lambda_{tc} \ll \lambda_{ct}$ for which the beating of the *cis* sets the beating of the *trans* and is not influenced by the *trans*. It should be mentioned that

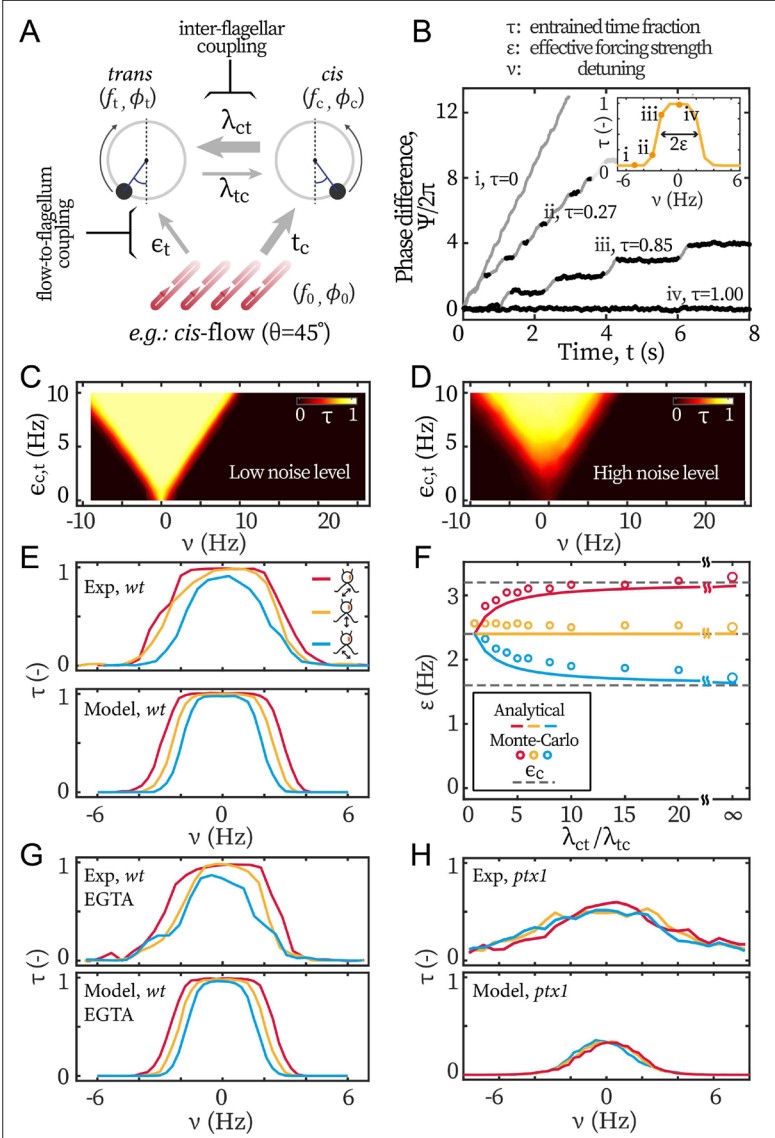

**Figure 5.** Modeling the asymmetric flow entrainment. (**A**) Modeling scheme describing a cell beating under directional flow (*cis*-flow as an example). Arrows represent the directional coupling coefficients with line thickness representing the relative strength. For example, $\lambda_{ct}$ points from *cis* to *trans* and represents how sensitive the latter's phase ($\phi_t$) is to the former's phase ($\phi_c$). Meanwhile, the arrow of $\lambda_{ct}$ being thicker than $\lambda_{tc}$ means that $\phi_t$ is much more sensitive to $\phi_c$ than the other way around. $\epsilon_{c,t}$ denote the sensitivity of $\phi_{c,t}$ to the flow's phase $\phi_0$. (**B**) Modeled phase dynamics of flow entrainment under frontal-flows, analogous to **Figure 2A**. Reproducing the Arnold tongue diagrams at the noise level of *wt* (**C**) and *ptx1* (**D**), analogous to **Figure 2B** and **Figure 2C** respectively. (**E**) Flow entrainment profiles $\tau(\nu; \theta)$ obtained experimentally (upper panel) and by modeling (lower panel). (**F**) $\epsilon(\theta)$ as a function of the inter-flagellar coupling asymmetry $\lambda_{ct}/\lambda_{tc}$. Points: measured from simulation; lines: analytical approximation (**Equation 3**); dashed lines: $\epsilon_c$ respectively for the *cis*-flow, frontal-flow, and *trans*-flow (from top to bottom). (**G**) Reproducing the flow entrainment of *wt* cells under calcium depletion. (**H**) Reproducing results of *ptx1*. See **Table 1** for the modeling parameters.

our main observation about the measured entrainment strength $\varepsilon$ only depending on the *cis*-loading, can also be reproduced by assuming $\epsilon_t \approx 0$. This alternative limit implies that the *trans* flagellum has no susceptibility to hydrodynamic loads, which is inconsistent with our entrainment experiments of *ptx*1. In addition, $\alpha \approx 1$ is also necessary to explain the difference in noise and effective temperature between the *wt* and *ptx*1 experiments.

**Table 1.** Modeling parameters.

| variable | symbol (unit) | TRIS | EGTA | *ptx1* |
|---|---|---|---|---|
| Intrinsic freq. *Kamiya and Hasegawa, 1987*; *Okita et al., 2005* | $f_c, f_t$ (Hz) | 45,65 | 45,65 | 45,65 |
| Basal coupling* | $\lambda_{ct} + \lambda_{tc}$ (Hz) | 60 | 60 | 60 |
| *cis* dominance *Okita et al., 2005*; *Horst and Witman, 1993* | $\lambda_{ct} : \lambda_{tc}$ (-) | 4:1 | 4:1 | 1:1 |
| Flow detuning | $\nu$ (Hz) | [−10,10] | [−10,10] | [−10,10] |
| Total forcing *Klindt et al., 2016* | $\epsilon_c + \epsilon_t$ (Hz) | 4.8 | 4.08 | 4.8 |
| Noise† *Quaranta et al., 2015* | $T_c, T_t$(rad²/s) | 1.57, 9.42 | 1.57, 9.42 | 9.42, 9.42 |

*detailed in Appendix 1: Monte-Carlo simulations.
†detailed in *Appendix 1: Noise in the beating of the ptx1 mutant.*

We perform Monte-Carlo simulations to solve the stochastic *Equation 2*, and determine the parameters of our simulations that reproduce our experimental results.

## Coordinated beating under symmetric forcing

The values of the parameters used in our simulations are summarized in *Table 1*. We first model the flow entrainment induced by frontal-flow, for which the flagella are loaded symmetrically and $\epsilon_c = \epsilon_t$.

We take $f_c = 45$ Hz and $f_t = 65$ Hz following *Kamiya and Hasegawa, 1987*; *Okita et al., 2005*. We set $\epsilon_{c,t}$ as 2.4 Hz to match the measured $\varepsilon$(frontal-flow) from *Figure 3A*. The noises for *cis* and *trans* flagellum are $T_c = 1.57$ rad²/s and $T_t = 9.42$ rad²/s respectively. At similar detunings as in the experimental results in *Figure 2A*, our Monte-Carlo simulations reproduce the phase dynamics with: (*i*) no flow entrainment, (*ii-iii*) unstable entrainment, and (*iv*) stable entrainment (*Figure 5B*). Repeating the simulations for varying forcing strengths $\epsilon_{c,t}$ and frequencies $f_0$ yields Arnold tongue diagrams in agreement with those reported from our experiments. For *wt*, we assume asymmetric coupling strength, $\lambda_{ct} = 4\lambda_{tc}$, while we assume a symmetric coupling strength $\lambda_{ct} = \lambda_{tc}$ for *ptx1*. With these values, the Arnold Tongue for *wt* in *Figure 2B* and *ptx1* in *Figure 2C* are reproduced with simulations shown in *Figure 5C and D* respectively. For *wt*, the asymmetry $\lambda_{ct} \gg \lambda_{tc}$ does not affect the overall shape of the Arnold tongue but leads to a low noise level, which is induced by the *cis*, and is much lower than the noise of *ptx1*.

## Coordinated beating under selective loading

Next, we model flow entrainment by the *cis*-flows and the *trans*-flows. The selective forcing ($\epsilon_c \neq \epsilon_t$) allows the effect of flagellar dominance ($\lambda_{ct} \neq \lambda_{tc}$) to manifest in the effective forcing strength $\varepsilon$ and hence in the entrainment profiles $\tau(\nu)$, *Figure 5E*. We derive the ratio between $\epsilon_c$ and $\epsilon_t$ from our computations, see *Figure 1E–G*, such that $\epsilon_c : \epsilon_t = 2:1$, 1:1, and 1:2 for the *cis*-flows, the frontal-flows, and the *trans*-flows respectively. With this setting, the model reproduces the experimental observations that the entrainment profile of *cis*-flow is consistently broader than that of frontal-flow (i.e. larger $\varepsilon$), and the profile of *trans*-flow is always the narrowest (smallest $\varepsilon$), see *Figure 5E*. *Figure 5F* shows how the asymmetry of inter-flagellar coupling ($\lambda_{ct} / \lambda_{tc}$) affects the asymmetry between the entrainment strength $\varepsilon$ for the *cis*-flows and the *trans*-flows. The open symbols represent $\varepsilon$ measured from modeled entrainment profiles $\tau(\nu)$ and the lines represent *Equation 3*. The difference between $\varepsilon$(*cis*-flow) and $\varepsilon$(*trans*-flow) increases with $\lambda_{ct} / \lambda_{tc}$. At large $\lambda_{ct} / \lambda_{tc}$, the $\varepsilon$ saturates to the forcing on the *cis* flagellum ($\epsilon_c$, see the grey dashed lines in *Figure 5F*).

Our experimental results for *wt* cells under calcium depletion are reproduced with a lower total forcing strength (*Figure 5G*). $\epsilon_c + \epsilon_t$ is set to 4.08 Hz (15% lower) to reflect the 7% − 20% decrease in $\varepsilon$ induced by calcium depletion (*Figure 3E*).

To reproduce the entrainment profiles of *ptx1* in *Figure 5H*, both a stronger noise and a symmetric inter-flagellar coupling are needed: while the stronger noise lowers the maximal values of $\tau(\theta, \nu)$, setting $\lambda_{ct} / \lambda_{tc} = 4$ would still result in $\tau$(*cis*-flow) > $\tau$(*trans*-flow) in the central range ($|\nu| \lesssim 2.4$ Hz).

Finally, it is noteworthy that the noise in *ptx1* increases not only because of a higher noise value for individual flagella, but also because the *cis-trans* coupling has become symmetric. As shown by *Equation 3*, the unilateral coupling promotes not only the *cis*-frequency in the synchrony but also the *cis*-noise. Given $T_c \ll T_t$ and $\lambda_{ct} = 4\lambda_{tc}$, we confirm with simulations that, for *wt*, the *cis* stabilizes the beating frequency of the *trans*. The simulations are in good agreement with experimental noise measurements, see *Appendix 1: Noise in the beating of the ptx1 mutant* for details.

## Discussion

The two flagella of *C. reinhardtii* have long been known to have inherently different dynamic properties such as frequency, waveform, level of active noise, and responses to second messengers (*Kamiya and Hasegawa, 1987*; *Okita et al., 2005*; *Leptos et al., 2013*; *Wan, 2018*; *Saegusa and Yoshimura, 2015*). Intriguingly, when connected by basal fibers and beating synchronously, they both adopt the kinematics of the *cis*-(eyespot) flagellum, which led to the assumption that the flagella may have differential roles in coordination. In this work, we test this hypothesis by employing oscillatory flows applied from an angle with respect to the cells' symmetry axis and thus exert biased loads on one flagellum.

In all our experiments with *wt* cells, we robustly observe that *cis*-flows, the ones that selectively load the *cis* flagellum, are always more effective in entraining the flagellar beating than the *trans*-flows. This is shown by the larger effective forcing strengths ($\varepsilon(cis\text{-flow}) > \varepsilon(trans\text{-flow})$, *Figure 3B*) and larger entrained time fractions ($\tau(cis\text{-flow}) > \tau(trans\text{-flow})$, *Figure 3C*). Mapping $\varepsilon(\theta)$ as a function of the loads, we find empirically that the flow entrainment strength scales with the hydrodynamic load on the *cis*, $\varepsilon \propto \overline{F}_{\text{Flow}}^{\text{c}}$ (*Figure 3D*) and that *trans*-loads appear to matter negligibly. These observations all indicate that the *cis*-loads determine whether an external forcing can entrain the cell. Moreover, this point is further highlighted by an unexpected finding: when *trans*-flows are applied, the *trans* flagellum always beats against the external flow ($P_{\text{Flow}}^{\text{t}} < 0$) and the only stabilizing factor for flow entrainment is the *cis* flagellum working transiently along with the flow ($P_{\text{Flow}}^{\text{c}} > 0$) during the recovery stroke (*Figure 1G* lower panel). These observations definitively prove that the two flagella have differential roles in the coordination and interestingly imply that flagella are coupled to external flow only through the *cis*.

To elucidate the mechanisms at the origin of this asymmetry, we develop a reduced stochastic model for the system, see *Equation 2*. In the model, selective hydrodynamic loading and flagellar dominance in the coordinated beating are respectively represented by $\epsilon_c \neq \epsilon_t$ and $\lambda_{ct} \neq \lambda_{tc}$. Using this model, we express $f$ and $\varepsilon$, which we can measure experimentally (*Equation 3*), as a function of $f_{c,t}$, $\epsilon_{c,t}$, $\lambda_{ct}$ and $\lambda_{tc}$ to illustrate how the flagellar dominance and selective loading affect the coordinated flagellar beating. Moreover, with Monte-Carlo simulations, we clarify the interplay between flows and flagella (*Appendix 1: Monte-Carlo simulations*), and reproduce all experimental observations. We show that a 'dominance' of the *cis* ($\lambda_{ct} \gg \lambda_{tc}$) is sufficient to explain the experimental phenomenology comprehensively. This dominance means that the *cis*-phase is much less sensitive to the *trans*-phase than the other way around. We then reproduce the phase dynamics of flow entrainment at varying detunings (*Figure 5B*), amplitudes (*Figure 5C*), and noises (*Figure 5D*). Exploiting the observation that the coordination between flagella cannot be broken by external flows up to the strongest ones tested ($\varepsilon^{\text{max}} \sim 10$ Hz, *Figure 2B*), we quantify the lower limit of the total basal coupling, $\lambda_{ct} + \lambda_{tc}$, to be approximately 40 Hz (deduced in *Appendix 1: Monte-Carlo simulations*), which is an order magnitude larger than the hydrodynamic inter-flagellar coupling (*Quaranta et al., 2015*; *Brumley et al., 2014*; *Klindt et al., 2016*; *Pelliciotta et al., 2020*).

Dynamic modulation of flagellar dominance during synchronous beating is the basis of *C. reinhardtii*'s tactic motility (*Kamiya and Witman, 1984*; *Horst and Witman, 1993*; *Pazour et al., 2005*; *Okita et al., 2005*). Calcium has long been speculated to facilitates such dynamic modulation (*Rüffer and Nultsch, 1997*; *Wan et al., 2014*; *Guo et al., 2021*) because it is involved in switching the beating mode (*Hayashi et al., 1998*) and in tuning flagellar beating amplitude (*Kamiya and Witman, 1984*; *Okita et al., 2005*), and calcium influx comprises the initial step of the cell's photo- (*Harz and Hegemann, 1991*) and mechanoresponses (*Yoshimura, 2011*). We therefore investigate flagellar coupling in the context of tactic steering by depleting the environmental free calcium and hence inhibiting signals of calcium influxes. Cells are first acclimated to calcium depletion, and then tested with the directional flows. Our results show that the *cis* dominance in the synchronous beating does not require

the involvement of free environmental calcium. Calcium depletion merely induces an overall drop in the forcing strength perceived by the cell $\varepsilon(\theta)$ (7% − 20%), which is captured by reducing $\epsilon_c + \epsilon_t$ for 15% (mean drop) in the model (*Figure 5G*). In contrast to the speculation that calcium is involved in the dynamic modulation of ciliary dominance in synchronous beating (*Rüffer and Nultsch, 1997*; *Wan et al., 2014*; *Guo et al., 2021*), our results indicate that the leading role of *cis* (i.e., $\lambda_{ct} \gg \lambda_{tc}$) is an inherent property, which does not require active influx of external calcium, and possibly reflects an intrinsic mechanical asymmetry of the cellular mesh that anchors the two flagella into the cell body.

In *ptx1* cells, a lack of flagellar dominance ($\lambda_{ct} = \lambda_{tc}$) and a stronger noise level are necessary to reproduce our experimental observations. Previous studies suggested that both flagella of *ptx1* are similar to the wildtype *trans* (*Horst and Witman, 1993*; *Rüffer and Nultsch, 1997*; *Okita et al., 2005*), and that the noise levels of this mutant's synchronous beating are much greater than those of *wt* (*Leptos et al., 2013*) (see also *Appendix 1: Noise in the beating of the ptx1 mutant*). If both flagella and their anchoring roots indeed have the composition of the wildtype *trans*, such symmetry would predict $\lambda_{ct} = \lambda_{tc}$. This symmetric coupling leads to a noise for *ptx1* $T \approx T_t$ (*Equation 3*), which is about an order of magnitude larger than the noise of *wt* $T \approx T_c$.

The comparison between *ptx1* and *wt* highlights an intriguing advantage of the observed unilateral coupling ($\lambda_{ct} \gg \lambda_{tc}$); that is, it strongly suppresses the high noise of the *trans*. Considering that the *trans* is richer in CAH6 protein and this protein's possible role in inorganic carbon sensing (*Mackinder et al., 2017*; *Choi et al., 2016*), the *trans* may function as a more active sensor than the *cis*. Assuming the sensing-related bio-activities is at the origin of *trans*' strong noise, then because the unilateral coupling prevents the noise from disrupting the cell's synchronous beating and effective swimming, it allows the cell to combine the benefit of having a stable *cis* as the driver and a noisy *trans* as a sensor.

## Materials and methods

### Cell culture

*C. reinhardtii* wildtype (*wt*) strain cc125 (mt+) and flagellar dominance mutant *ptx1* cc2894 (mt+), obtained from the Chlamydomonas Resource Center, are cultured in TRIS-minimal medium (pH = 7.0) with sterile air bubbling, in a 14 hr/10 hr day-night cycle. Experiments are performed on the fourth day after inoculating the liquid culture, when the culture is still in the exponential growth phase and has a concentration of ~2 × 10$^5$ cells/ml. Before experiments, cells are collected and resuspended in fresh TRIS-minimal (pH = 7.0).

### Calcium depletion

In calcium depletion assays, cells are cultured in the same fashion as mentioned above but washed and resuspended in fresh TRIS-minimal medium +0.5 mM EGTA (pH = 7.0). Free calcium concentration is estimated to drop from 0.33 mM in the TRIS-minimal medium, to 0.01 µM in the altered medium (*Wakabayashi et al., 2009*). Experiments start at least one hour after the resuspension in order to acclimate the cells.

### Experimental setup

Single cells of *C. reinhardtii* are studied following a protocol similar to the one described in *Quaranta et al., 2015*. Cell suspensions are filled into a customized flow chamber with an opening on one side. The air-water interface on that side is pinned on all edges and is sealed with silicone oil. A micropipette held by micro-manipulator (SYS-HS6, WPI) enters the chamber and captures single cells by aspiration. The manipulator and the captured cell remain stationary in the lab frame of reference, while the flow chamber and the fluid therein are oscillated by a piezoelectric stage (Nano-Drive, Mad City Labs), such that external flows are applied to the cell. Frequencies and amplitudes of the oscillations are individually calibrated by tracking micro-beads in the chamber. Bright field microscopy is performed on an inverted microscope (Nikon Eclipse Ti-U, 60× water immersion objective). Videos are recorded with a sCMOS camera (LaVision PCO.edge) at 600–1000 Hz.

### Measurement scheme

The flagellar beating of each tested cell is recorded before, during, and after the application of the flows. We measure the cell's average beating frequency $f$ over 2 s (~100 beats). For *ptx1* cells, $f$ is

reported for the in-phase (IP) synchronous beating. Unless otherwise stated, directional flows (θ = 0°, 45°, 90°, 135°) are of the same amplitude (780±50 μm/s, mean±std), similar to those used in *Quaranta et al., 2015*. Flow frequencies $f_0$ are scanned over $[f - 7, f + 7]$ Hz for each group of directional flows.

## Computation of the flagellar loads

To quantify the hydrodynamic forces on the flagella, we first track realistic flagellar deformation from videos wherein background flows are applied. Then we employ a hybrid method combining boundary element method (BEM) and slender-body theory (*Keller and Rubinow, 1976*; *Wei et al., 2021*) to compute the drag forces exerted on each flagellum and the forces' rates of work. In this approach, each flagellum is represented as a slender-body (*Keller and Rubinow, 1976*) with 26 discrete points along its centerline and the time-dependent velocities of each of the 26 points are calculated by the point's displacement across frames. The cell body and the pipette used to capture the cell are represented as one entity with a completed double layer boundary integral equation (*Power and Miranda, 1987*). Stresslet are distributed on cell-pipette's surface; while stokeslets and rotlets of the completion flow are distributed along cell-pipette's centerline (*Keaveny and Shelley, 2011*). The no-slip boundary condition on the cell-pipette surface is satisfied at collocation points. Lastly, stokeslets are distributed along the centerlines of the flagella, so that no-slip boundary conditions are met on their surfaces. Integrating the distribution of stokeslets $\mathbf{f}(s)$ over a flagellar shape, one obtains the total drag force $\mathbf{F} = \int \mathbf{f}(s)ds$. Similarly, the force's rate of work is computed as $P = \int \mathbf{f}(s) \cdot \mathbf{U}(s)ds$, where $\mathbf{U}(s)$ is the velocity of the flagellum at the position $s$ along the centerline.

The computations shown in this study are based on videos of a representative cell which originally beats at ~50 Hz. The cell is fully entrained by flows along different directions (θ = 0°, 45°, 90°, 135°) at 49.2 Hz. In the computations, the applied flows are set to have an amplitude of 780 μm/s to reflect the experiments. Computations begin with the onset of the background flows (notified experimentally by a flashlight event), and last for ~30 beats (500 frames sampled at 801 fps). Additionally, we confirm the results of *trans*-flow-entrainment, that both flagella spend large fractions of time beating against the flows, with other cells and with *trans*-flows at other frequencies.

## Isolate loads of external flows

The total loads ($F$ and $P$) computed consist of two parts, one from the flow created by the two flagella themselves and the other from the flow imposed. In the low Reynolds number regime, the loads of the two parts add up directly (linearity): $\mathbf{F} = \mathbf{F}_{\text{Self}} + \mathbf{F}_{\text{Flow}}$, and $P = P_{\text{Self}} + P_{\text{Flow}}$. To isolate $\mathbf{F}_{\text{Flow}}$ and $P_{\text{Flow}}$, we compute $\mathbf{F}' = \mathbf{F}_{\text{Self}}$ and $P' = P_{\text{Self}}$ by running the computation again but without the external flows, and obtain $\mathbf{F}_{\text{Flow}} = \mathbf{F} - \mathbf{F}'$ and $P_{\text{Flow}} = P - P'$.

## Compute time fraction of phase-locking

In practice, phase-locking is considered established if phase difference, either between the two flagella (Δ) or between the flow and flagellum (Ψ), varies slow enough over time. Here, we use Ψ to illustrate the process. We first break down an entire time series ($t_{\text{tot}} \approx 10$ s) to segments of 0.1 s (~5 beats). A given segment is considered phase-locked if $\left|\frac{d\Psi}{dt}\right| \leq \pi$ rad/s. This particular threshold ($\pi$ rad/s) is equivalent to a frequency mismatch of 0.5 Hz, which is smaller than our frequency resolution in scanning the detuning (~0.8 Hz).

## Modeling parameters

We assume the flagellar intrinsic frequencies $f_c$ and $f_t$ to be 45 Hz and 65 Hz respectively (*Kamiya and Hasegawa, 1987*; *Okita et al., 2005*; *Wan et al., 2014*). On this basis, $\lambda_{ct} : \lambda_{tc}$ is assumed to be 4:1 to account for the observed $f \sim 50$ Hz. $\epsilon_c : \epsilon_t$ is set as 2:1, 1:1, and 1:2 for the *cis*-flows, the frontal-flows, and the *trans*-flows respectively, see *Figure 1E–G*. The underlying assumptions are: the two flagella are equally susceptible to loads, and the effective coupling strength is linearly proportional to the hydrodynamic load. Additionally, $\epsilon_c + \epsilon_t$ is assumed to be constant to reflect the fact that $\overline{F}_{\text{Flow}}^c + \overline{F}_{\text{Flow}}^t$ does not vary with flow directions (0.74–0.79 $F_{\text{swim}}$). The noise levels for the *cis*- and the *trans* flagella are taken as $T_c, T_t = 1.57, 9.42$ rad²/s. Under unilateral flagellar coupling, the collective noise level approximates to $T_c$ and the value corresponds to typical experimental observations (*Quaranta et al., 2015*). The sum of inter-flagellar coupling $\lambda = \lambda_{ct} + \lambda_{tc}$ is set to be large enough, i.e., $\lambda = 3 |f_t - f_c|$, to

account for the fact that: (1) the coordinated beating is approximated in-phase, and (2) up until the strongest flow applied, the inter-flagellar synchronization cannot be broken (quantitative evaluation is detailed in *Appendix 1: Monte-Carlo simulations*). To model *wt* cells under calcium depletion, we decrease $\epsilon_c + \epsilon_t$ by 15% - which is the mean decrease in the observed $\varepsilon(cis\text{-flow})$, $\varepsilon(frontal\text{-flow})$, and $\varepsilon(trans\text{-flow})$(*Figure 3E*). For *ptx1* cells, we assume a symmetric inter-flagellar coupling and set the noise level of both flagella to that of the *trans*, $T_t = 9.42$ rad$^2$/s. The parameters are summarized in *Table 1*.

## Acknowledgements

The authors thank Roland Kieffer for technical support. DW thanks Ritsu Kamiya for helpful discussions. The authors acknowledge support by the European Research Council (ERC starting grants no. 716712 and no. 101042612, and ERC consolidator grant no. 101045302).

## Additional information

### Funding

| Funder | Grant reference number | Author |
| --- | --- | --- |
| European Research Council | 101042612 | Marie-Eve Aubin-Tam |
| European Research Council | 716712 | Daniel SW Tam |
| European Research Council | 101045302 | Daniel SW Tam |

The funders had no role in study design, data collection and interpretation, or the decision to submit the work for publication.

### Author contributions

Da Wei, Conceptualization, Formal analysis, Investigation, Methodology, Writing – original draft, Writing – review and editing; Greta Quaranta, Investigation, Methodology; Marie-Eve Aubin-Tam, Conceptualization, Resources, Supervision, Funding acquisition, Investigation, Project administration, Writing – review and editing; Daniel SW Tam, Conceptualization, Resources, Formal analysis, Supervision, Funding acquisition, Investigation, Methodology, Writing – review and editing

### Author ORCIDs

Da Wei ⬤ https://orcid.org/0000-0002-6226-0639
Marie-Eve Aubin-Tam ⬤ http://orcid.org/0000-0001-9995-2623
Daniel SW Tam ⬤ https://orcid.org/0000-0001-5300-0889

### Decision letter and Author response

Decision letter https://doi.org/10.7554/eLife.86102.sa1
Author response https://doi.org/10.7554/eLife.86102.sa2

## Additional files

### Supplementary files

• MDAR checklist

### Data availability

The data and the code underlying this study is openly available at the 4TU. ResearchData repository: https://doi.org/10.4121/21972695. Codes for modeling and plotting, and the hydrodynamic computations' data are included.

The following dataset was generated:

| Author(s) | Year | Dataset title | Dataset URL | Database and Identifier |
|---|---|---|---|---|
| Wei D, Quaranta G, Aubin-Tam M-E, Tam DSW | 2023 | Dataset and codes underlying "The younger flagellum sets the beat for *C. reinhardtii*" | https://doi.org/10.4121/21972695 | 4TU.ResearchData, 10.4121/21972695 |

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

# Appendix 1

## Hydrodynamic computation for asymmetric loading

Here, we present the computation results for the flagellar loads under flows of different directions, see *Appendix 1—figure 1*. Realistic flagellar shapes are tracked from the video recordings. Background flows begin with a marked event (flash light) which defines the zero time (see the dashed lines at t=0). After several beats (shaded in blue), the cell becomes entrained by the flows, see the left panels *Appendix 1—figure 1A1-H1*. Meanwhile, the right panels (A2-H2) show the median and the interquartile range of $F_{Flow}$ and $P_{Flow}$ over the flow-entrained beats, respectively. Force magnitudes are scaled by $F_{swim} = 6\pi\mu R U_{swim} = 9.9$ pN, which is the Stokes drag on a typical free-swimming cell (radius $R$=5 μm, swim velocity $U_{swim}$ = 110 μm/s); while the viscous powers are scaled by $P_{swim} = F_{swim}U_{swim} = 6\pi\mu R U_{swim}^2 = 1.1$ fW. Here $\mu$ = 0.95 mPa·s is the dynamic viscosity of water at 22 °C. Panels A2-F2 are presented and described in the main text. For the flow along 0°, the mean force is 0.37 $F_{swim}$ and 0.34 $F_{swim}$ (panel G2) while the mean power is –0.2 $P_{swim}$ and –0.4 $P_{swim}$ (H2), for the *cis* and the *trans* flagellum respectively.

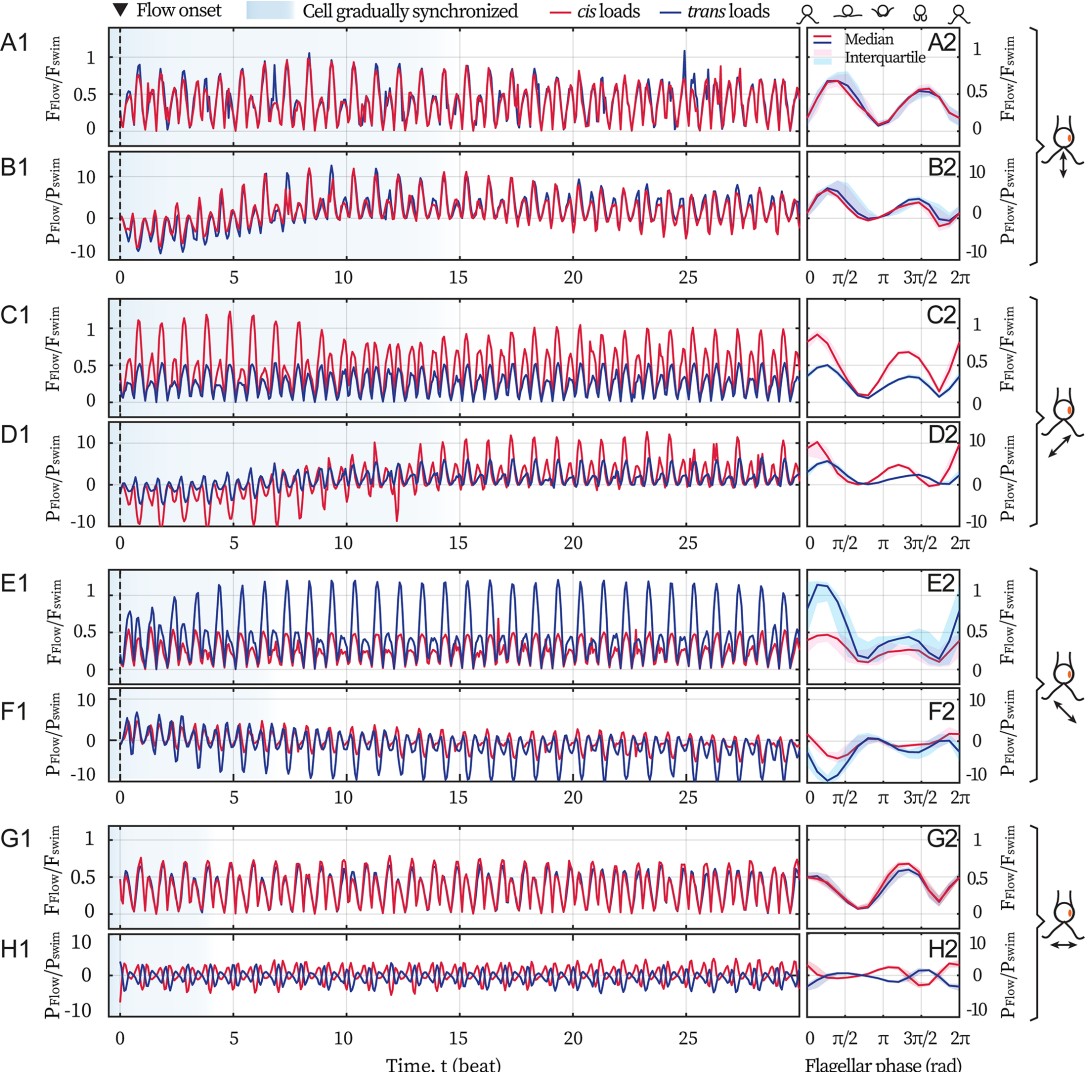

**Appendix 1—figure 1.** Hydrodynamic computations of the loads on the flagella. Computations are based on recordings of different directional flows (49.2 Hz) entraining the same cell ($f \approx 50$ Hz). Computed drag forces (**A1**) and the force's rate of work (**B1**) on the *cis* (red) and the *trans* (blue) flagellum for 30 cycles, when the cell is subjected to the frontal-flow ($\theta$ = 90°). Black dashed lines mark the onset of the applied flow. Blue shading in the background represents the beating cycles before the cell is entrained. Magnitude of the drag forces $F_{Flow}$ (**A2**) and the power of the flow $P_{Flow}$ (**B2**) averaged over the flow-entrained cycles. The solid lines and the shaded

*Appendix 1—figure 1 continued on next page*

*Appendix 1—figure 1 continued*

areas represent the median and the interquartile range respectively. (**C–D**) Same as (**A–B**) but for the *cis*-flow ($\theta$ = 45°). (**E–F**) Results for the *trans*-flow ($\theta$ = 135°). (**G–H**) Results for the flow with $\theta$ = 0°. Scaling factors $F_{\text{swim}}$=9.9 pN and $P_{\text{swim}}$=1.1 fW.

## Extracting coupling strength by fitting phase dynamics

In the main text, the flagellum-flow forcing strength $\varepsilon$ in *wt* cells is mainly extracted from the entrainment profile ($\tau(\nu) \geq 0.5$). Meanwhile, in previous works (**Polin et al., 2009**; **Quaranta et al., 2015**), fitting the distribution of phase dynamics is employed to extract $\varepsilon$. In the latter approach, the idea is that the phase locking during entrainment leads to a peaked probability distribution of $\Psi$ (the difference between the flagellar phase and the flow's phase), whose width is broadened by the noise. The distribution, $P(\Psi)$, can be derived from the Adler equation (**Equation 1** in the main text) as:

$$P(\Psi) = \int_{\Psi}^{\Psi+2\pi} \exp\left(\frac{V(\Psi') - V(\Psi)}{T}\right) d\Psi'. \tag{4}$$

Here $V(\Psi) = -2\pi\nu\Psi - 2\pi\varepsilon\cos(\Psi)$ is a wash-board potential, $T$ is the effective temperature in phase dynamics.

Here, we demonstrate that these two approaches are equivalent in extracting $\varepsilon$. For all *wt* cells tested in the TRIS-minimal medium (N=11), their $\varepsilon$ measured by the $\tau(\nu)$ width and extracted from fitting are plotted against each other, **Appendix 1—figure 2**. All points center around the identity line (y=x), showing the equivalence in obtaining $\varepsilon$ by the two methods. For the *ptx1* dataset displayed in the main text, $\varepsilon$ are extracted from fitting the phase dynamics.

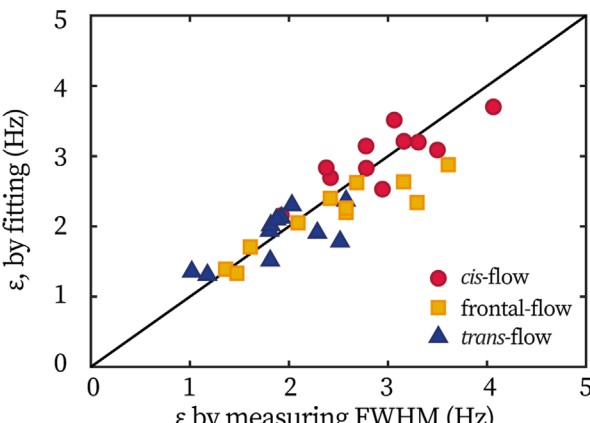

**Appendix 1—figure 2.** Equivalence of extracting coupling strength $\varepsilon$ by different methods. Each point represents one cell under either the frontal-flow (yellow square), the *cis*-flow (red circle), or the *trans*-flow (blue triangle). The $x$ coordinate is the coupling strength $\varepsilon$ measured by the half width of entrainment profile, $\tau(\nu) \geq 0.5$; and the $y$ coordinate is obtained by fitting the flagellar phase dynamics.

## Monte-Carlo simulations

### Governing equations for the phase dynamics

The external flow and the two flagella are described by three coupled ordinary differential equations (ODEs), for the phase of the *cis* flagellum $\phi_c$, the phase of the *trans* flagellum $\phi_t$, and the phase of the external flow $\phi_0$:

$$\begin{cases} \dot{\phi}_0 = 2\pi f_0 \\ \dot{\phi}_c = 2\pi f_c - 2\pi\lambda_{tc}\sin(\phi_c - \phi_t) - 2\pi\epsilon_c\sin(\phi_c - \phi_0) + \zeta_c \\ \dot{\phi}_t = 2\pi f_t - 2\pi\lambda_{ct}\sin(\phi_t - \phi_c) - 2\pi\epsilon_t\sin(\phi_t - \phi_0) + \zeta_t. \end{cases} \tag{5}$$

The phase dynamics $\phi_0$, $\phi_c$ and $\phi_t$ depend on the intrinsic frequencies, with $f_0$ the frequency of the external flow, $f_c$ and $f_t$ the intrinsic beating frequencies of the *cis* and the *trans* flagellum respectively. $\lambda_{tc}$ represents the strength of the forcing exerted by the *trans* flagellum on the *cis* flagellum and $\lambda_{ct}$ the forcing exerted by the *cis* flagellum on the *trans* flagellum. Finally, $\epsilon_c$ represents the strength of the susceptibility of the *cis* flagellum to flow entrainment and $\epsilon_t$ the susceptibility of the *trans* flagellum to flow entrainment. The flow is assumed to be noise free (*Equation 5*), the *cis* and the *trans*-flagella are assumed to have noise levels $\zeta_c$ and $\zeta_t$ respectively. The noises are assumed to be Gaussian, such that $\langle \zeta_{c,t}(\tau + t)\zeta_{c,t}(\tau)\rangle = 2\,T_{c,t}\delta(t)$. In this study, the phase dynamics of these equations are examined using Monte-Carlo simulation. The temporal resolution of the simulations ($dt$) is 1ms, which corresponds to the experimental frame rates (801 Hz). Hereafter, we derive analytical solutions to these equations in the limits of oscillatory steady states. For this, we rewrite the phase dynamics as functions of the inter-flagellar phase difference $\Delta$ and the average flagellar phase $\Phi$ defined as:

$$
\begin{cases}
\Delta = (\phi_c - \phi_t), \\
\Phi = (\phi_c + \phi_t)/2, \\
\Psi = \Phi - \phi_0.
\end{cases}
\tag{6}
$$

Here, $\Psi$ represents the difference between the average phase $\Phi$ and the phase of the flow forcing $\phi_0$. For a complete summary of the symbols used throughout this article, please see *Appendix 1—table 1*. *Equation 5* can be rewritten as:

$$
\begin{cases}
\dot{\Delta} = 2\pi\left[\nu_{ct} - \lambda\sin\Delta - \epsilon_c\sin\left(\Psi + \dfrac{\Delta}{2}\right) + \epsilon_t\sin\left(\Psi - \dfrac{\Delta}{2}\right)\right] + \zeta_c - \zeta_t, \\
\dot{\Psi} + \dfrac{\lambda_{ct} - \lambda_{tc}}{2\lambda}\dot{\Delta} = 2\pi\left[-\nu - \dfrac{\lambda_{ct}\epsilon_c}{\lambda}\sin\left(\Psi + \dfrac{\Delta}{2}\right) - \dfrac{\lambda_{tc}\epsilon_t}{\lambda}\sin\left(\Psi - \dfrac{\Delta}{2}\right)\right] + \zeta,
\end{cases}
\tag{7}
$$

where $\nu_{ct} = f_c - f_t$, $\lambda = \lambda_{ct} + \lambda_{tc}$, $\nu = f_0 - f$, $f = \dfrac{\lambda_{ct}f_c + \lambda_{tc}f_t}{\lambda}$, $\zeta = \dfrac{\lambda_{ct}\zeta_c + \lambda_{tc}\zeta_t}{\lambda}$. The phases from both flagella can be easily recovered from the phase difference and the average phase, as: $\phi_c = \Phi + \Delta/2$, $\phi_t = \Phi - \Delta/2$.

**Appendix 1—table 1.** Summary of the symbols used in both the main text and the Appendix 1.

| | |
|---|---|
| $f$ | Frequency of the synchronous beating without external flow |
| $f_{c,t}$ | Intrinsic frequencies of the *cis* or *trans* flagellum |
| $f_0$ | Frequency of the external flow |
| $\nu$ | Detuning, the flow-flagella frequency mismatch, $f_0 - f$ |
| $\phi$ | Flagellar phase |
| $\phi_{c,t}$ | Phase of the *cis* or *trans* flagellum |
| $\phi_0$ | Phase of the external flow |
| $\Delta$ | Phase difference between flagella, $\phi_c - \phi_t$ |
| $\Phi$ | Mean flagellar phase $(\phi_c + \phi_t)/2$ |
| $\Psi$ | Phase difference between the flow and the flagella, $\Phi - \phi_0$ |
| $\varepsilon$ | Effective forcing strength, describing flow-flagella coupling |
| $\epsilon_{c,t}$ | Forcing strength of external flow on the *cis* or *trans* flagellum |
| $\lambda_{ct}$ | Influence of the *cis* flagellar phase on the *trans* flagellar phase |
| $\lambda_{tc}$ | Influence of the *trans*-phase on the *cis*-phase |

*Appendix 1—table 1 Continued on next page*

Appendix 1—table 1 Continued

| $f$ | Frequency of the synchronous beating without external flow |
|---|---|
| $\zeta$ | Noise of the synchronous beating |
| $\zeta_{c,t}$ | Noise of the *cis* or *trans* flagellum |
| $T$ | Effective temperature of the synchronous beating |
| $T_{c,t}$ | Effective temperature of the *cis* or *trans* flagellum |

## Flagellar synchronization

We first consider the synchronization between the *cis-* and the *trans* flagella, in the absence of external flow, such that we assume: $\epsilon_c = 0$ and $\epsilon_t = 0$. When the flagella are synchronized to one another, $\dot{\Delta} = 0$, we can directly deduce from *Equation 7* that the two flagella beat synchronously at the frequency:

$$f = \frac{\lambda_{ct} f_c + \lambda_{tc} f_t}{\lambda}. \tag{8}$$

In addition, from *Equation 7* we can deduce the steady-state phase difference $\Delta$, which satisfies:

$$\sin \Delta = \frac{\nu_{ct}}{\lambda}. \tag{9}$$

It is therefore obvious that the two flagella can only synchronize ($\dot{\phi}_{c,t} = f$) when $\left| \nu_{ct} / \lambda \right| \leq 1$.

## Interaction between three oscillators

We focus our discussion on the case when the *cis* and the *trans* flagella are synchronized in phase, as is observed in all of our experiments. Here, $\dot{\Delta} = 0$ and $\Delta \approx 0$, which is realized for values of the parameters such that: $\lambda \gg \epsilon_{c,t} \gg \nu_{ct}$. *Equation 7* reduces to:

$$\dot{\Psi} = 2\pi \left[ -\nu - \varepsilon \sin \Psi \right] + \zeta, \tag{10}$$

with $\varepsilon = \left( \lambda_{ct} \epsilon_c + \lambda_{tc} \epsilon_t \right) / \lambda$. For the sake of clarity, we recall here that $\nu = f_0 - f$, $f = \left( \lambda_{ct} f_c + \lambda_{tc} f_t \right) / \lambda$, $\lambda = \lambda_{ct} + \lambda_{tc}$ and $\zeta = \left( \lambda_{ct} \zeta_c + \lambda_{tc} \zeta_t \right) / \lambda$. It is noteworthy that *Equation 10* is the same Adler equation as the one introduced in *Equation 1* of the main text. This equation represents the phase dynamics of a single phase oscillator evolving under external periodic forcing (*Pikovsky et al., 2001*). In this strong-coupling limit, when inter-flagellar coupling strength $\lambda$ are much larger than flow's effective forcing strength $\varepsilon$, the coupled flagella simply work as one single oscillator of intrinsic frequency $f$. The phase dynamics is characterized by $f$, $\varepsilon$ and $\zeta$, which are weighted averages of $f_{c,t}$, $\varepsilon_{c,t}$ and $\zeta_{c,t}$ - the characteristics of the *cis* and *trans* flagella. Our main experimental observations can all be explained by the limit when the inter-flagellar forcing of the *cis* on the *trans* is much larger than that of the *trans* on the *cis*, such that $\lambda_{ct} \gg \lambda_{tc}$. In this case, $\varepsilon \approx \epsilon_c$, which is consistent with the observation that flow entrainment is only a function of the hydrodynamic loads on the *cis* flagellum. Lastly, $\zeta \approx \zeta_c$ is consistent with the observation that the noise in the phase dynamics when *wt* cells are entrained by the flow, is much smaller than the noise recorded for *ptx1*, which is putatively considered to have two *trans* flagella. In fact, when $\lambda_{ct} \gg \lambda_{tc}$, $\zeta \approx \zeta_c$ satisfies even in the absence of imposed flows, which explains why the breaststroke beating (IP mode) of *ptx1* is much noisier than that of *wt* (see the last section of Appendix 1).

## Lower limit of inter-flagellar coupling

Lastly, we turn to the model's behaviors when two flagella are not necessarily in synchrony, and we focus on the regime where $\lambda - \left| \nu_{ct} \right|$ is comparable to $\epsilon_{c,t}$. The primary question is: from the fact that even the strongest flow in our experiment could not disrupt inter-flagellar synchrony, can we infer a lower limit of $\lambda$? We assume $\left| \nu_{ct} \right| = 20$ Hz (*Kamiya and Hasegawa, 1987*; *Kamiya, 2000*; *Okita et al., 2005*; *Wan et al., 2014*) and focus on the entrainment by the frontal-flow. To further simplify the picture, coupling between *cis* and the *trans* are set as equal, $\lambda_{ct} = \lambda_{tc}$.

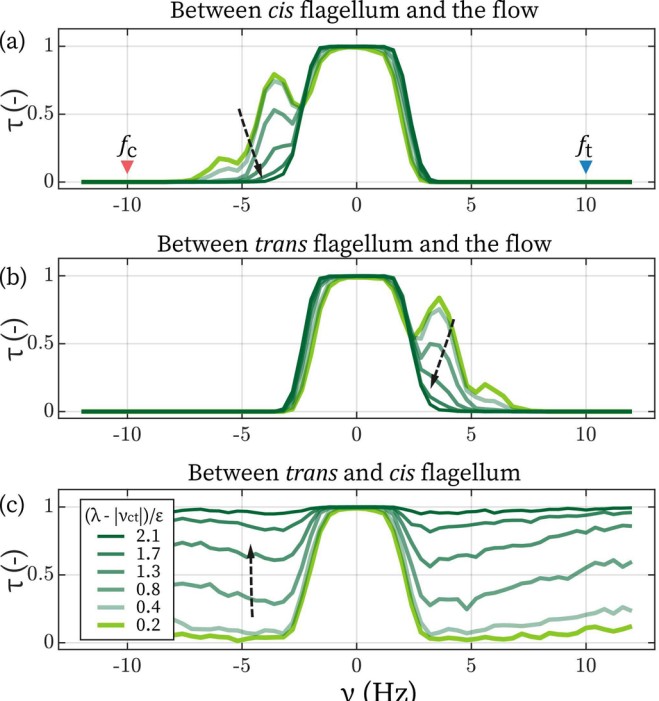

**Appendix 1—figure 3.** Determine the lower limit of $\lambda$. The time fractions of the *cis* (**a**) and the *trans* flagellum (**b**) entrained by the flow. (**c**) The time fraction of where *cis* and *trans* are synchronized. Arrows points towards increasing $\left(\lambda - \left|\nu_{\mathrm{ct}}\right|\right)/\varepsilon$.

The value $\left(\lambda - \left|\nu_{\mathrm{ct}}\right|\right)/\varepsilon$ with $\varepsilon = \epsilon_{c,t}$ determines if the flow can disrupt the synchrony between *cis* and *trans*. We plot the synchronization/entrainment profiles with increasing $\lambda$ in ***Appendix 1—figure 3***. When it satisfies $\left(\lambda - \left|\nu_{\mathrm{ct}}\right|\right)/\varepsilon \geq 2$, external flows cease to affect the flagellar synchronization observably. As the strongest flow ($21 U_{\mathrm{swim}}$) applied experimentally corresponds to $\varepsilon \approx 10$ Hz, altogether, we conclude that $\lambda \gtrsim \left|\nu_{\mathrm{ct}}\right| + 2\varepsilon_{\max} = 40$ Hz. In the main text, we set $\lambda = 3\left|\nu_{\mathrm{ct}}\right| = 60$ Hz, which satisfies this relation and matches the observation that the phase lag between the flagella ($\Delta$) is small.

## Noise in the beating of the *ptx1* mutant

Here, we show an as-yet uncharacterized strong noise present in the synchronous beating of the mutant *ptx1*. The in-phase (IP) mode of *ptx1* cells and the breaststroke beating of the *wt* cells are similar in waveform and frequency (**Horst and Witman, 1993**; **Rüffer and Nultsch, 1997**; **Leptos et al., 2013**). However, the former has a much stronger noise.

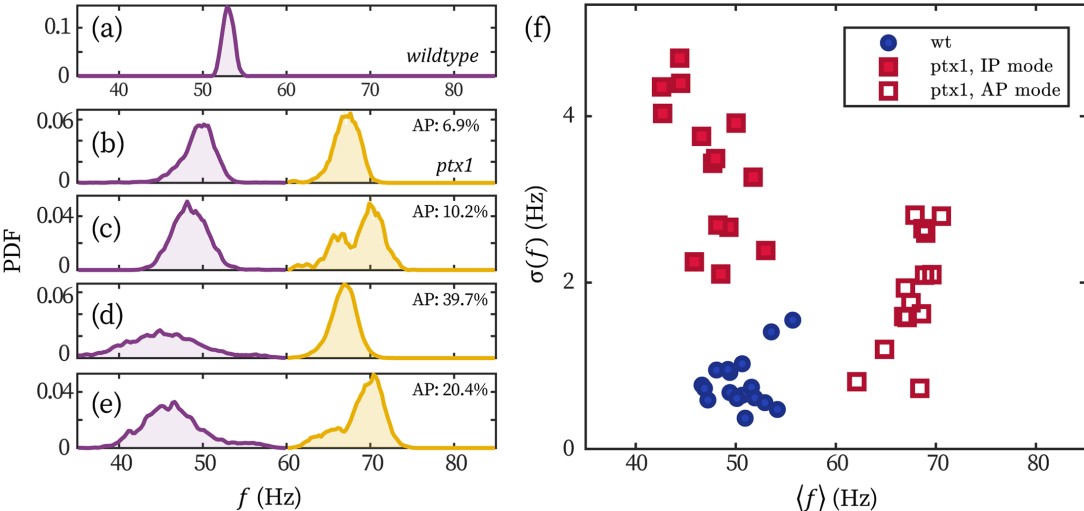

**Appendix 1—figure 4.** Stronger frequency fluctuation of the IP mode of *ptx1* cells. (**a–e**) Representative probability distributions of the beating frequency of a *wt* (**a**) and four *ptx1* cells (**b–e**) over 30 seconds. Probability distributions of the IP (purple) and AP mode (yellow) are respectively normalized for better visualization. The time fractions of the AP mode are noted in each panel. (**f**) The *wt* and *ptx1* cells represented by its mean beating frequency $\langle f \rangle$ and the standard deviation of the beating frequencies over time $\sigma(f)$.

The strong noises show obviously in fluctuations of IP beating frequencies (similar to data shown in *Leptos et al., 2013*). In *Appendix 1—figure 4*, we display the distribution of beating frequency of a representative *wt* cell (panel a) and four representative *ptx1* cells (panels b-e). The broad peaks of the IP (purple) and AP (yellow) beating of *ptx1* sharply contrast the narrow peak of *wt*. We quantify the frequency fluctuations of all the cells displayed in the main text (N=11 for *wt* and N=14 for *ptx1*) in *Appendix 1—figure 4f*. The cells are represented by their mean beating frequencies over time $\langle f \rangle$ and the standard deviations $\sigma(f)$. Clearly, the breaststroke beating of *wt*, the IP, and the AP mode of *ptx1* each forms a cluster. The *wt* cluster is at $(\langle f \rangle, \sigma(f)) = (50.5 \pm 2.6, \ 0.8 \pm 0.3)$ Hz (mean±1 std. over cell population); and is less dispersed than both the IP- and the AP modes of *ptx1*, which are at $(47.4 \pm 3.1, \ 3.4 \pm 0.9)$ Hz and $(67.6 \pm 2.1, \ 1.9 \pm 0.7)$ Hz, respectively. Under the assumption of a white (Gaussian) noise, $\sigma(f)$ is proportional to the amplitude of phase noise $\zeta$, and thus scales with $\sqrt{T}$. Considering that $\sigma(f)$ for *ptx1* is three- to fivefolds larger than that of *wt*, we conclude that the effective temperature of phase dynamics in *ptx1* is an order of magnitude larger than *wt*, $T^{ptx1}/T^{wt} \sim \mathcal{O}(10)$.

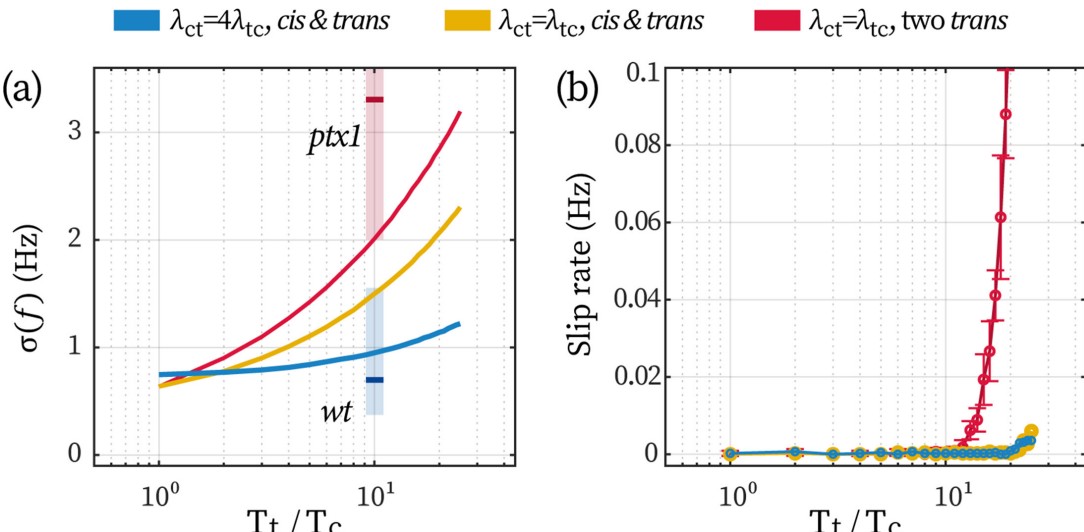

**Appendix 1—figure 5.** Effect of a low-noise *cis* in stabilizing the beating of the *trans*. (**a**) Fluctuations in beating frequency ($\sigma(f)$) under different coupling schemes and flagellar noises. Other model parameters are the same as used in the main text. The colors of the lines correspond to the three cases explained in the text hereafter. The red and blue vertical shades represent the experimentally observed range for *ptx1* and *wt* cells, respectively, with short bars marking the mean values. (**b**) The rate of slip under the conditions. Error bars correspond to 1 std. over N=9 repetitions.

The stronger noise in *ptx1* can be attributed to two sources, namely, the loss of a stable *cis* and the loss of the unilateral coupling, **Appendix 1—figure 5**. We perform Monte-Carlo simulations of the coupled beating of two flagella under three conditions (see the blue, yellow, and red data in **Appendix 1—figure 5** respectively): (1) a *cis* coupled with a *trans* unilaterally ($\lambda_{ct} = 4\lambda_{tc}$); (2) a *cis* coupled with a *trans* reciprocally ($\lambda_{ct} = \lambda_{tc}$); and (3) two *trans* flagella coupled with each other reciprocally. We fix the noise level of the *cis* at $T_c = 1.57\ \mathrm{rad}/s^2$ and we vary the noise of the *trans* to compare how the increase in trans-noise would destabilize the synchronous flagellar beating.

When a *trans* is unilaterally following the *cis*, increasing the *trans*-noise over an order of magnitude only leads to a ~20% stronger frequency fluctuation (the blue line in **Appendix 1—figure 5a**). On the contrary, lacking either the unilateral coupling or the low-noised *cis* would increase the fluctuation for 200% (yellow line) or 300% (red line). Qualitatively, simulation results are in agreement with experimental measurements assuming that $T_t/T_c \sim \mathcal{O}(10)$, see the red and blue shaded areas in **Appendix 1—figure 5a**. Additionally, even when the *cis*- and *trans* flagella exert equal forcing on one another ($\lambda_{ct} = \lambda_{tc}$), having a low-noise *cis* flagellum already helps prevent phase slips from interrupting the inter-flagellar synchrony. This can be seen from **Appendix 1—figure 5b**: as long as a low-noise *cis* is present (yellow and blue data), slips will be sparse (< 0.01 Hz). Together, these simulation results highlight the stabilizing effect of a low-noise *cis* flagellum, and illustrate that a unilateral coupling further enhances the stabilization.

