## [Editor Report]

This important manuscript investigates how the two flagella of *C. reinhardtii*, which have inherently different dynamic properties, synchronize their beating. Through careful, non-invasive experiments, the authors help the field to understand the mechanisms responsible for this synchronization, arguing that these mechanisms are due to unilateral coupling between the two flagella. The data are convincing, and the conclusion about unilateral coupling has been strengthened by additional analysis during revision.

---

## [Decision Letter]

**Decision letter after peer review:**

Thank you for submitting your article "The younger flagellum coordinates the beating in *C. reinhardtii*" for consideration by *eLife*. Your article has been reviewed by 3 peer reviewers, one of whom is a member of our Board of Reviewing Editors, and the evaluation has been overseen by Anna Akhmanova as the Senior Editor.

Essential revisions (for the authors):

The referees unanimously agreed that the experiments are original and the results of the experiments are interesting. The referees also raised significant concerns that should be addressed:

1) Clarity of the writing: the referees found the manuscript difficult to read, especially to a life-science reader, which could potentially diminish the impact of the work on this community. The authors are encouraged to rethink the structure, organization, and writing of their results.

2) Models assumptions: the referees pointed out the opacity of the assumptions that entered in the synchronization model. The authors are invited to revise their presentation of the model to clarify the underlying assumptions and explain the relationship between Equation (1) and Equation (2).

3) Main conclusion: the manuscript's main conclusion is that the elegant experimental results can be explained by a simple model with unilateral coupling between the two flagella. However, the referees were not convinced that unilateral coupling is proven, because the analysis presented in the manuscript does not rule out that the observations could be explained in terms of different susceptibilities between the two flagella. The authors are invited to qualify their main conclusion in light of this discussion.

*Reviewer #1 (Recommendations for the authors):*

The manuscript investigates how the cis-(eyespot) and trans flagella of *C. reinhardtii* synchronize their beating. The two flagella are known to have inherently different dynamic properties such as frequency, waveform, level of active noise, and responses to second messengers, but, in intact cells, they are connected by basal fibers and beating synchronously, with both adopting the kinematics of the cis flagellum. It was thus hypothesized that the flagella have differential roles in coordination. Here, the authors test this hypothesis by non-invasively employing oscillatory flows that are applied at an angle with respect to the cells' symmetry axis, and thus exert biased loads on one flagellum. The authors show that the coordinated beating responds to only load exerted on the cis flagellum. The authors argue that this asymmetry in response derives from a unilateral coupling between the two flagella.

The study is interesting and novel; it addresses an important question in cilia coordination. The experiments are carefully conducted, and the analysis is convincing. The findings, that the cis flagellum dominates flagellar coordination and coupling to external flows, are intriguing and valuable.

I was confused when I first read the manuscript between the different uses of the word synchronization – synchronization among the flagella versus synchronization of each flagellum with the superimposed oscillatory flow. I think the manuscript should be edited to avoid such confusion.

*Reviewer #2 (Recommendations for the authors):*

The main strength of the paper is that the experiments are imaginative, the analysis and modeling rigorous and the conclusions appear to be solid.

The main weakness is that the paper is challenging to read and will be impenetrable to the biological readership. The elliptic presentation of results precludes assessment of the strength of the conclusions.

1. The symbol epsilon may be being used with different meanings (e.g., Equation 1, Figure 1F inset: is synchronization of the cis or trans with the force?). Please explain why they are the same, if indeed they are.

2. It is essential to show more data so that the reader can understand what's going on. For example, in Figure 1F, also show phi_c – phi_t (this important data).

3. Figures 1B-F are not referenced until after Figure 2, making it impossible for the reader to follow the argument.

4. Like comment 2 above, more data needs to be included in Figure 2: show both F_total and F_self because, without more explanation, it is very difficult to grasp this result.

5. It would be useful have a "Results" section and for the sub-headers to be sentences (with verbs) describing the findings.

6. Figures should be labeled using text, not just symbols (some of which may not be defined such as tau(-) on the axis of Figure 3B).

7. "Theta_c-flow" (and many other examples) is a clumsy compound noun. The authors should consider descriptors such as "frontal" for theta_a and "rightward" for theta_c etc. (see point 8 below)

8. I suggest rotating the little diagrams in Figure 1A so that front is upwards and they agree with the diagrams between Figure 2A and 2B.

9. A main conclusion is that they use the modeling to back out λ_c/λ_f. The authors state a value of 4, but could they give a range of values that are consistent with their data?

I suggest the authors ask some biology graduate students or employ a professional writer to help them make this important paper comprehensible.

*Reviewer #3 (Recommendations for the authors):Chlamydomonas reinhardtii* is a biflagellate microalga widely employed as a model system to study eukaryotic cilia. Its two flagella, termed cis and trans, are physiologically different and mechanically coupled intracellularly by fibres. The authors use an experimental system that was successfully employed in the past to study flagellar entrainment, to analyse the differences that emerge when forcing cis and trans flagella differently. This is achieved by holding a single cell with a micropipette and periodically oscillating the background fluid along the flagellar plane but at an angle with respect to the axis of the cell. A flow comes from the cis-side of the cell loads the cis flagellum more than the trans. The symmetric case loads the trans flagellum more than the cis. The results are compared to front loading, to forcing with a calcium chelator (as calcium is known to affect flagellar dynamics differently for cis and trans) and to the phototaxis mutant ptx1. This mutant is currently understood to have both flagella of the trans type. When looking at the flagellar pair as a single oscillator, the authors find that the entrainment results can always be modelled as a noisy Adler model, commonly used for phase oscillators. The authors then link this to a simplified model where the flagella are separately represented by self sustained phase oscillators with non-reciprocal coupling and independent couplings to the external flow. This is used to conceptualise the main results of the paper and in particular the idea that the cis flagellum acts as a stabiliser of the coupled dynamics.Strengths:

1) The idea behind the paper is interesting and novel.

2) The experimental results are of high quality. The tests on the potential role of calcium and on the differences with flagellar mutants (here ptx1) provide interesting complementary information. They also represent a good example of meaningful interdisciplinarity between physics and biology.

3) The use of a minimal model for the forced dynamics of the flagellar pair gives a powerful intuitive grasp of the essential physics of the phenomenon.

Weaknesses:

1) The relation between the single flagellar oscillator model (Equation 1) and the two flagellar oscillators model (Equation 2) is insufficiently developed. The result is that it is not clear whether there is enough evidence to agree with the authors' interpretation of the differential loading experiments for the wild type strain (see private recommendations).

2) There should be more information on how the loads on the flagella are calculated. Changing the forcing, the waveforms will likely change and it is these modified waveforms that need to be taken into account to calculate the loads. The Materials and methods hint a the fact that the waveforms used were those of a representative cell. However, there is no information on whether these waveforms were obtained for a cell without loading or whether different waveforms were obtained for different loadings.

Below I summarise the main suggestions I have for the authors.

A) My main concern with this manuscript is in the link between Equation 1 and Equation 2, in particular in regards to the parameters \epsilon_c,t(\theta_c,t). These parameters represent the phase susceptibility of the cis and trans phase oscillators to the external loading. The manuscript estimates the loading from the flows and the waveforms, which broadly speaking I agree with (but see specific point on this below). However, the parameter \epsilon is not the forcing. Rather, it characterises *the phase response* of the flagellum to a certain forcing. It will be linked to the forcing only through a relation which is currently unknown and certainly quite complex. Starting from the estimated forcing curves, the manuscript assumes that \epsilon_c(\theta_a)=\epsilon_t(\theta_a), \epsilon_c(\theta_c)=\epsilon_t(\theta_t), \epsilon_c(\theta_t)=\epsilon_t(\theta_c). However, given the difference between forcing and susceptibility, I do not see a compelling evidence in support of these relations. If cis and trans flagella have different susceptibility to external forcing, these relations will not be true. Even \epsilon_c(\theta_a) does not need in principle to be the same as \epsilon_t(\theta_a).

The effective parameters (Equation 1) are functions of all of the intrinsic parameters (Equation 2) and it seems to me that there is enough redundancy to obtain the same experimental phenomenology with differences in the \epsilon's rather than in the \λ's.

In my view this is an important issue because it underpins the main conclusion of the paper. The authors should explore this point in more detail.

B) I find the discussion deriving the Equations S6 and S7 unsatisfactory. Whilst I can agree to consider \δ slaved to the variable \phi0-\phi_f, I do not necessarily agree with the "small \δ" approximation that brings S6 to S7. I agree that the variable \δ can be small, but here the question is whether it is much smaller than (\phi0-\phi_f). This I do not think is necessarily clear a priori. For \δ \sim (\phi0-\phi_f), Equation S7 is not the correct approximation to S6. Notably, the parameters of the resulting equation for d\phi0/dt will depend also e.g. on (\epsilon_c-\epsilon_t). I think the authors should work out this section much more explicitly and justify better their approximations.

C) In the phase oscillator model of Equations S2, the two flagella are assumed to "have the same level of noise (\zeta_c=\zeta_t)" and at the same time their variances show different effective temperatures. I think that this needs to be clarified as the two things do not seem to be compatible to me. (Also, I think there is a typo in Equations S3. The phases on the left hand sides are inverted).

D) Throughout the paper the intrinsic frequencies of cis and trans are fixed to f_c=45Hz and f_t=65Hz. However, these parameters will certainly vary from cell to cell. I think it would be good to justify why we can forget about this variability in the analysis. It is not clear to me.

E) If I understand correctly, the load estimates are done with a single waveform, derived from a sample cell. Upon load, the waveforms will change. These (changed) waveforms are the ones that should be used to determine the load. Is this the case in this manuscript? From the Materials and methods this doesn't seem to be the case.

F) The manuscript states that "trans-loads appear to matter negligibly". This needs to be discussed in more detail, given the fact that the ptx1 flagella, which are putatively both trans, can be entrained by the hydrodynamic loads quite well.

[Editors' note: further revisions were suggested prior to acceptance, as described below.]

Thank you for resubmitting your work entitled "The younger flagellum sets the beat for *C. reinhardtii*" for further consideration by *eLife*. Your revised article has been evaluated by Anna Akhmanova (Senior Editor), a guest Reviewing Editor, and two reviewers.

The manuscript has been improved but there are some remaining issues that need to be addressed, as outlined below:

*Reviewer #3 (Recommendations for the authors):*

I very much appreciate the efforts made by the authors in improving their manuscript. I think the current version is better organised and reads better as well. As I wrote in my previous report, I find their results interesting and valuable and I think this paper should be published.

Still, there are a couple of final comments that I would like to put to the authors. I am particularly interested in them addressing point (3) below. The other two are of secondary importance with respect to this one.

(1) It is still unclear to me how \tau is calculated and what is its relation with the parameters of the Adler model. Isn't there an analytical expression (or at least semi-analytical) that can be put forward? Deciding that \epsilon can be measured from the region \tau\geq0.5 seems a bit arbitrary to me. This boundary seems to work well for the 11 cells tested (see Appendix), but can the method be extrapolated to the general case, e.g. when there is an arbitrary value of the beating noise? I would expect that keeping \epsilon but changing T would decrease "t_entrain" and therefore the width of the region \tau\geq0.5.

(2) Please provide more details on how t_entrain is calculated.

(3) On pgs. 15,16 the model is compared to the experimental data. Figure 5E shows that starting with \epsilon equal to the experimental value of the frontal flow (2.4Hz) and detuning the frequency of the forcing, f_0, one obtains a synchronisation curve with a width of 2.4Hz. Was this not a given to begin with? I must be missing something. The width of the synchronisation region is precisely given by the value of \epsilon, so I do not understand how this could have given anything different. The same goes for Figure 5C,D, where it is clear that the boundaries of the Arnold's tongue are straight lines at 45 degrees.

The experiments have been analysed in terms of an Adler model. The parameters are then inserted in a model whose effective dynamics is equivalent to that of an Adler model. We should then expect to recover the behaviour of the experiments. This seems to me to be a bit of a circular argument and I do not understand how the results of Figure 5 could have been any different. I think it would be very helpful if the authors could clarify this point in the paper.

---

## [Author Response]

Essential revisions (for the authors):The referees unanimously agreed that the experiments are original and the results of the experiments are interesting. The referees also raised significant concerns that should be addressed:(1) Clarity of the writing: the referees found the manuscript difficult to read, especially to a life-science reader, which could potentially diminish the impact of the work on this community. The authors are encouraged to rethink the structure, organization, and writing of their results.

We have reorganized the manuscript and rewritten a considerable fraction of it. The major revisions include: (1) a complete reorganization of the manuscript, with a separate result section including an improved description of our main results and sub-headers summarizing the findings, (2) a new version of figures 1, 2, 3 and 4 with panels reorganized in response to the Reviewers’ concerns and the text modified accordingly, (3) a clearer derivation of our model using new notations, labels and symbols to improve the readability, which we summarize in the Table presented in Appendix 1 Section Monte-Carlo Simulations. Meaning of the symbols are denoted in the figures and are explained in more detail, (4) the terminologies are updated to avoid confusion.

(2) Models assumptions: the referees pointed out the opacity of the assumptions that entered in the synchronization model. The authors are invited to revise their presentation of the model to clarify the underlying assumptions and explain the relationship between Equation (1) and Equation (2).

The modeling section has been significantly revised to enhance its clarity. The changes in notations and structure of the manuscript improve the readability and clarify the derivation of the model. The assumptions are now explicitly stated in both the Model section (main text) and in the Material and Methods section. How the assumption (strong inter-flagellar coupling) connects Equation 1 (Adler equation) and Equation 2 (our model) is introduced in the paragraph starting at Line#268.

(3) Main conclusion: the manuscript's main conclusion is that the elegant experimental results can be explained by a simple model with unilateral coupling between the two flagella. However, the referees were not convinced that unilateral coupling is proven, because the analysis presented in the manuscript does not rule out that the observations could be explained in terms of different susceptibilities between the two flagella. The authors are invited to qualify their main conclusion in light of this discussion.

The new manuscript directly discusses our main conclusion in the context of the effect of different susceptibilities between the two flagella, which was not clearly presented in the original manuscript.

We have reorganized the manuscript to discuss our results on the *ptx1* mutant towards the beginning of the result section. These experiments are relevant to this discussion because both flagella of *ptx1* are thought to be *trans* flagella. In our results, the mutant displays similar susceptibility to flow synchronization (entrainment) as *wt*, and we show that *trans* is at least equally (and possibly more) susceptible to hydrodynamic loads. This observation is in line with previous results on flagellar load response (see Klindt et al., PRL, 2016 and response to Reviewer 3’s comment A). This discussion justifies our modeling assumption that the susceptibilities of *trans* and *cis* are comparable.

Two additional observations further support the unilateral coupling assumption we adopt in our model. (1) The synchronous beating frequency of the *cis-trans* pair is closer to the intrinsic frequency of *cis* flagellum and (2) the noise level observed in *wt* is much lower than a noisy *trans* flagellum would induce if the inter-flagellar coupling is reciprocal. These observations are not explained by different susceptibilities between the flagella, but are comprehensively captured by the unilateral coupling assumption.

We have explicitly discussed the possibility of differential flagellar susceptibility and justify our conclusion of unilateral coupling, see for example the text starting from Line#164 and #274.

Reviewer #1 (Recommendations for the authors):The manuscript investigates how the cis-(eyespot) and trans flagella of *C. reinhardtii* synchronize their beating. The two flagella are known to have inherently different dynamic properties such as frequency, waveform, level of active noise, and responses to second messengers, but, in intact cells, they are connected by basal fibers and beating synchronously, with both adopting the kinematics of the cis flagellum. It was thus hypothesized that the flagella have differential roles in coordination. Here, the authors test this hypothesis by non-invasively employing oscillatory flows that are applied at an angle with respect to the cells' symmetry axis, and thus exert biased loads on one flagellum. The authors show that the coordinated beating responds to only load exerted on the cis flagellum. The authors argue that this asymmetry in response derives from a unilateral coupling between the two flagella.The study is interesting and novel; it addresses an important question in cilia coordination. The experiments are carefully conducted, and the analysis is convincing. The findings, that the cis flagellum dominates flagellar coordination and coupling to external flows, are intriguing and valuable.I was confused when I first read the manuscript between the different uses of the word synchronization – synchronization among the flagella versus synchronization of each flagellum with the superimposed oscillatory flow. I think the manuscript should be edited to avoid such confusion.

We would like to thank Reviewer 1 for the helpful suggestion. We have updated the manuscript to make a clear distinction between these two different synchronizations. The term “synchronization” is now mostly used to refer to the synchronization between the two flagella, and we use the term “entrainment” for the synchronization of the flagella with the external flow. The term “entrainment” is used in the literature for the synchronization with an external stimulus [e.g., “entrainment of respiration by a mechanical ventilator”, Pikovsky et al. Chap. 3 in the book Synchronization (2001)] and implies a unidirectional influence, *i.e.,* the flow forces the flagellum but not the other way around.

The readers are explicitly reminded about the terminology:

Line#81: “Hereafter, we refer to the synchronization of the flagellar beating with the imposed external flow as “flow entrainment”, in order to avoid confusion with the “synchronization” between the cis and the trans flagella.”

Reviewer #2 (Recommendations for the authors):The main strength of the paper is that the experiments are imaginative, the analysis and modeling rigorous and the conclusions appear to be solid.The main weakness is that the paper is challenging to read and will be impenetrable to the biological readership. The elliptic presentation of results precludes assessment of the strength of the conclusions.

We thank Reviewer 2 for the thorough review of our manuscript and for the constructive suggestions. We have reorganized the manuscript and rewritten most of the text. The Reviewer’s comments have improved the readability of the manuscript and we hope the revised manuscript is now more welcoming to the biological readership.

1. The symbol epsilon may be being used with different meanings (e.g., Equation 1, Figure 1F inset: is synchronization of the cis or trans with the force?). Please explain why they are the same, if indeed they are.

We have clarified the definition of ε, ϵc, and ϵt throughout the manuscript. First, we want to highlight that, in our study, the two flagella always beat in synchrony (ϕc=ϕt). Hence, the *cis* and the *trans* flagella are phase-locked, and observed to beat as a single oscillator (see also our response to the second comment).

The symbol ε (without subscript) denotes the effective forcing strength of external flow on the two flagella – which beat as *one* oscillator under all flow conditions (frontal-flow, cis-flow and trans-flow). The coupling strength ε is measured experimentally for each flow condition, as the width of the detuning range wherein flow entrainment is established (see Figure 2A). On the other hand, the symbols ϵc and ϵt denote the forcing exerted by the flow on the *cis* and *trans* flagellum, respectively (also highlighted in Figure 5A). They differ from each other in the different flow conditions and characterize the asymmetric loading that we impose. Note that ϵc and ϵt are not directly observed and are only used in the model (see Equation 2). Our model allows us to infer how ε depends on both ϵc and ϵt, see Equation 3. We have clarified this difference throughout the manuscript:

Line#270: “From this equation, we can directly write the quantities f, ε and T measured in our experiments as a function of the parameters from our model fc,t, ϵc,t, λct,tc, and ζc,t”

Lastly, to better acquaint the readers about the meaning of ε (*e.g.*, why it is an “effective” forcing strength *etc.*), we explain Equation 1 in more details in the revised manuscript:

Line#134: “Or more precisely, ε describes how sensitive is the flagellar phase to the external stimuli, and it thus depends on both the absolute strength of the flow as well as the susceptibility of the flagella.”

Line#139: “By definition, entrainment corresponds to a steady-state (time-invariant) solution of Ψ. Solving Equation 1 with Ψ˙=0, one sees that entrainment is only possible when the effective forcing strength is strong enough: |ε|>|ν|. In this light, we can experimentally scan ν and measure ε directly by the frequency range where flow entrainment can establish.”

Line 146: “[…(the) range in ν for which τ≥0.5] is considered as the region where entrainment establishes, and is used as a measure for the flagellum-flow coupling strength ε.”

2. It is essential to show more data so that the reader can understand what's going on. For example, in Figure 1F, also show phi_c – phi_t (this important data).

The phase difference (ϕc−ϕt) shown in the inset of Figure 1F in the original manuscript is now presented in the new Figure 1, as an inset in Figure 1D, which shows Δ=ϕc−ϕt fluctuating around 0 when there is no external flow. As mentioned in the previous response, *cis* and *trans* flagellum always beat synchronously, with or without the external flow. Author response image 1 presents time series of Δ that correspond to trace *(i)* to *(iv)* in Figure 2.

**Author response image 1. sa2fig1:** 

We have clarified that *cis* and *trans* are always phase-locked (ϕc=ϕt) throughout the manuscript:Line#75: “In all our experimental recordings, the *cis* and the *trans* flagellum beat synchronously. Typical phase locking between them is represented in Figure 1D inset, where the phase difference Δ=ϕc−ϕt fluctuates around 0. Therefore, ϕc and ϕt are considered equal and denoted as ϕ.”

Line#83: “In our experiments, the *cis* and *trans* flagella always beat synchronously, therefore flow entrainment always takes place for both flagella simultaneously, regardless of the flow's direction.”

3. Figures 1B-F are not referenced until after Figure 2, making it impossible for the reader to follow the argument.

We thank the Reviewer for pointing this out. We have rewritten the manuscript and reorganized the figure panels in the resubmitted manuscript. In particular, the panels showing the experimental setup, flow-loading scheme, and signal analysis are now grouped as the new Figure 1. Thereafter, a new Figure 2 focuses on the flow entrainment under symmetric loading (frontal-flows), with which we introduce and explain the Adler equation, Equation 1.

4. Like comment 2 above, more data needs to be included in Figure 2: show both F_total and F_self because, without more explanation, it is very difficult to grasp this result.

We have now included a new figure and text description in Appendix 1 detailing the computational results (see the new Appendix 1 Section Hydrodynamic computation for asymmetric loading, and the new Appendix 1 figure 1). In the figure we present the magnitude of drag force on *cis* and *trans* flagellum, FFlowc,t, as a function of time, while the flagellar beatings are gradually entrained (synchronized) by external flows along different directions. We mention this in the main text:

Line#98: “Our computations provide the loads on the flagella throughout the entire experiment, from the time the captured cell is gradually entrained by the external flow, see Appendix 1: Hydrodynamic computation for asymmetric loading for the entire time series.”

The equality Ftotalc,t=Fselfc,t+FFlowc,t is a vector equality but the linearity will not be clear when plotting the magnitudes of Ftotalc,t and Fselfc,t. We have therefore opted not to show them in the manuscript and have instead clarified the manuscript:

Line#95: “In the linear Stokes regime, the drag force F is the sum of the contributions due to the motion of the flagella and the motion of external flow. Our simulations allow us to compute both contributions separately. We present the loads induced by the external flow FFlow and PFlow (see Methods for details).”

5. It would be useful have a "Results" section and for the sub-headers to be sentences (with verbs) describing the findings.

We have rewritten most of the manuscript and revised it accordingly.

6. Figures should be labeled using text, not just symbols (some of which may not be defined such as tau(-) on the axis of Figure 3B).

Thanks for the suggestion, we have now recognized the difficulty brought by the non-textual axis labels. In the updated version, labels of all axes are accompanied with textual definition within the figure. When space is the limiting factor, we place textual definition of the symbol in the vicinity of the panel (e.g., above Figure 3B or Figure 5B).

7. "Theta_c-flow" (and many other examples) is a clumsy compound noun. The authors should consider descriptors such as "frontal" for theta_a and "rightward" for theta_c etc. (see point 8 below)

The clumsy terms are replaced. We are now referring to the flows along the cell’s axis of symmetry as frontal-flows – as the Reviewer has suggested. After consideration, we now term the flow that selectively loads the *cis* (*trans*) flagellum as the *cis-*flow (*trans*-flow). The terms are clearly introduced in the first place, see Figure 1E-G.

Line 115: “Our computations demonstrate that flows along θ = 45° impose stronger loads on the cis flagellum, and we will refer to these as cis-flows, hereon forward. Likewise, flows on θ = 135° selectively load the trans and we denote these as trans-flows. Finally, the flows along 90 degrees that approach the cell from the front will be called frontal-flows.”

8. I suggest rotating the little diagrams in Figure 1A so that front is upwards and they agree with the diagrams between Figure 2A and 2B.

Changes are made accordingly. Now all schematic drawings of a *C. reinhardtii* cell point downward. We thank the Reviewer for the carefulness.

9. A main conclusion is that they use the modeling to back out λ_c/λ_f. The authors state a value of 4, but could they give a range of values that are consistent with their data?

For clarity, we replace the notation λc (and λt) as λct(andλtc) to specify the direction of influence between flagellum, e.g., λct mean how much *cis* influences *trans*.

Our major experimental finding (i.e., the cells’ asymmetric susceptibility to external flow entrainment) is captured as long as λct>λtc, this is shown in Figure 5F. Figure 5F clearly shows that for λct>4λtc, the values of ε for the different flows do not vary significantly, and the beating of the two flagella will be set by the cis-flagellum.

Line#307: “…details how the asymmetry of inter-flagellar coupling (λct/λtc) affects the asymmetry between the entrainment strength ε for the cis-flows and the trans-flows.”

The specific ratio λct/λtc=4 is deduced from the beating frequency f being closer to the *cis* frequency fc for typical frequencies reported in the literature. This can be seen from Equation 3, f=(λctfc+λtcft)λ with λ=λct+λtc. With intrinsic frequencies fc,t obtained from partial deflagellation experiments (wherein one flagellum is amputated and the other one’s frequency is regarded as its intrinsic frequency), fc=45 Hz, ft=65 Hz, and f≈ 50 Hz, we obtain λc/λt≈4. This is in line with in others’ data [see Figure 10 in Wan et al., J. R. Soc. Interface (2013), fc*=57.12 Hz,*
ft*=80.52 Hz and*
f0*=63.38 Hz,*
λc/λt≈4*.*]

I suggest the authors ask some biology graduate students or employ a professional writer to help them make this important paper comprehensible.

We thank the Reviewer for the suggestion. As described in the public response, we have made a major reorganization of the manuscript and rewritten large portions to make it more comprehensible.

Reviewer #3 (Recommendations for the authors):*Chlamydomonas reinhardtii* is a biflagellate microalga widely employed as a model system to study eukaryotic cilia. Its two flagella, termed cis and trans, are physiologically different and mechanically coupled intracellularly by fibres. The authors use an experimental system that was successfully employed in the past to study flagellar entrainment, to analyse the differences that emerge when forcing cis and trans flagella differently. This is achieved by holding a single cell with a micropipette and periodically oscillating the background fluid along the flagellar plane but at an angle with respect to the axis of the cell. A flow comes from the cis-side of the cell loads the cis flagellum more than the trans. The symmetric case loads the trans flagellum more than the cis. The results are compared to front loading, to forcing with a calcium chelator (as calcium is known to affect flagellar dynamics differently for cis and trans) and to the phototaxis mutant ptx1. This mutant is currently understood to have both flagella of the trans type. When looking at the flagellar pair as a single oscillator, the authors find that the entrainment results can always be modelled as a noisy Adler model, commonly used for phase oscillators. The authors then link this to a simplified model where the flagella are separately represented by self sustained phase oscillators with non-reciprocal coupling and independent couplings to the external flow. This is used to conceptualise the main results of the paper and in particular the idea that the cis flagellum acts as a stabiliser of the coupled dynamics.Strengths:1) The idea behind the paper is interesting and novel.2) The experimental results are of high quality. The tests on the potential role of calcium and on the differences with flagellar mutants (here ptx1) provide interesting complementary information. They also represent a good example of meaningful interdisciplinarity between physics and biology.3) The use of a minimal model for the forced dynamics of the flagellar pair gives a powerful intuitive grasp of the essential physics of the phenomenon.Weaknesses:1) The relation between the single flagellar oscillator model (Equation 1) and the two flagellar oscillators model (Equation 2) is insufficiently developed. The result is that it is not clear whether there is enough evidence to agree with the authors' interpretation of the differential loading experiments for the wild type strain (see private recommendations).2) There should be more information on how the loads on the flagella are calculated. Changing the forcing, the waveforms will likely change and it is these modified waveforms that need to be taken into account to calculate the loads. The Materials and methods hint a the fact that the waveforms used were those of a representative cell. However, there is no information on whether these waveforms were obtained for a cell without loading or whether different waveforms were obtained for different loadings.

We thank Reviewer 3 for the careful review and insightful comments. In the resubmitted manuscript and Appendix 1, we have addressed the concerns mentioned above (weakness #1-2).

To address weakness #1 (the unclarity of the model), the assumption (strong inter-flagellar coupling) that connects Equation 1 (Adler equation) and Equation 2 (our model) is now explicitly introduced (Line#268). The other possible interpretation of our experimental results i.e., differential flagellar susceptibility, as pointed out by the Reviewer in comment #A, is now openly discussed in Lines #164 and #279. In short, this possibility is ruled out, because both previous experiments in flagellar load responses and our results in the mutant *ptx1* show that the susceptibility of *cis* and *trans* flagellum are similar.

Additionally, we have made these following revisions to enhance the clarity and readability of the manuscript: (1) After reorganization, a new result section (Sec.1) now highlights that the susceptibility to flow synchronization (entrainment) of the *ptx1* and *wt* are comparable*.* (2) We have also significantly rewritten the modeling section. The model is now derived more clearly using new notations, labels, and symbols. The symbols are summarized in the Table presented in Appendix 1 Section Monte-Carlo simulations.

To address weakness #2, the necessity of more information about the hydrodynamic computations, we have given more details of the computation process (both in the Methodology section and in Method and Materials). We also present the entire time series of computed flow loads in the resubmitted Appendix 1 (Section Monte-Carlo simulations).

The Reviewer is also concerned about the accuracy of using a single flagellar waveform to compute loads. In fact, the computations in the manuscript use the flagellar waveforms directly (manually) extracted from the experimental videos, *i.e.*, the real-time waveform under each imposed oscillatory flow. We now emphasize this explicitly in the manuscript (Line#97) to avoid confusion.

Below I summarise the main suggestions I have for the authors.

Here we first summarize some changes in terminology – which we hope can be more self-explanatory and less ambiguous. The updated terms and symbols will be used in this reply, instead of the old ones.

In the revised manuscript, we now refer to the synchronization of the *cis-trans* pair with the flow as “entrainment” and the synchronization between *cis* and *trans* flagella “synchronization”. (Following suggestions from Reviewer 1)The names of the flows are now changed to reflect which flagellum they selectively load. Namely, θc-flow is now called the *cis*-flow and θt-flow the *trans*-flow. The θa-flow, or the axial flow, is now called the frontal-flow. (Following suggestions from Reviewer 2).The inter-flagellar coupling strengths, originally denoted as λc (λt) is now λct (λtc). The double subscripts clarify the direction of influence (illustrated in Figure 5A). The effective temperature, Teff now writes T for simplicity.

Below we reply to each suggestion of the Reviewer:

A) My main concern with this manuscript is in the link between Equation 1 and Equation 2, in particular in regards to the parameters \epsilon_c,t(\theta_c,t). These parameters represent the phase susceptibility of the cis and trans phase oscillators to the external loading. The manuscript estimates the loading from the flows and the waveforms, which broadly speaking I agree with (but see specific point on this below). However, the parameter \epsilon is not the forcing. Rather, it characterises *the phase response* of the flagellum to a certain forcing. It will be linked to the forcing only through a relation which is currently unknown and certainly quite complex. Starting from the estimated forcing curves, the manuscript assumes that \epsilon_c(\theta_a)=\epsilon_t(\theta_a), \epsilon_c(\theta_c)=\epsilon_t(\theta_t), \epsilon_c(\theta_t)=\epsilon_t(\theta_c). However, given the difference between forcing and susceptibility, I do not see a compelling evidence in support of these relations. If cis and trans flagella have different susceptibility to external forcing, these relations will not be true. Even \epsilon_c(\theta_a) does not need in principle to be the same as \epsilon_t(\theta_a).The effective parameters (Equation 1) are functions of all of the intrinsic parameters (Equation 2) and it seems to me that there is enough redundancy to obtain the same experimental phenomenology with differences in the \epsilon's rather than in the \λ's.In my view this is an important issue because it underpins the main conclusion of the paper. The authors should explore this point in more detail.

We thank the Reviewer for this important comment. The Reviewer points out the possibility of reproducing our experimental observation, including the asymmetric flow entrainment profiles (τ as a function of ν) with only asymmetry in the susceptibility of the *trans* and the *cis* flagellum to external flows (ϵc,t). This was not appropriately discussed in the original manuscript and we have added a discussion to address this point.

Indeed, this can be seen from Equation 3: ε=αϵc+(1−α)ϵt, with α=λct/(λct+λtc). The effective entrainment strength ε is equivalent to ϵc (i.e., the beat is set by the *cis* flagellum), when (1−α)=0 as we argue in the manuscript, or when the *trans* flagellum is not susceptible to the external loads, such that ϵt=0. Our conclusion was not sufficiently substantiated, and we have reorganized and partially rewritten the manuscript to better justify our conclusion that λct>λtc.

First, the assumption that the *trans* flagellum has much weaker sensitivity than the *cis* flagellum to external loads (ϵt∼0) is not supported by several experimental observations. As mentioned by the Reviewer in comment F, the flagella of the *ptx1* mutant are putatively both *trans* (see also Ref.23, 28, 38, 45, 46). Our results in Figure 2 (Arnold tongues in Figure 2B and 2C) show that the *ptx1 trans* flagella are at least as susceptible to external hydrodynamic loads as the *cis* and *trans* flagella of *wt*, therefore one cannot assume ϵt≪ϵc. This result is in qualitative agreement with another study of flow entrainment, which has characterized the load required to stall the beating of the flagella of *wt* (Ref.52). In this study, the authors find no significant difference in the load response of the *cis* and the *trans*, and note that near the stalling load, the *cis* can continue to beat while the *trans* flagellum is stalled by the hydrodynamic load (see video S1 in their supplementary material). This clearly indicates that the *trans* flagellum is at least as susceptible to external flows as the *cis* and possibly more.

Second, *trans* flagella have been reported to beat with larger noise levels (see Ref.22-27, 29 which report larger frequency fluctuations for the *trans*). Likewise, we observe that the noise level in the synchronous beating of the putative *trans* flagella in *ptx1* cells is much higher than that of the beating of *wt*. The low noise level measured for *wt* cells despite the higher noise level of their trans flagellum can be explained by Equation 3, T=α2Tc+(1−α2)Tt by assuming again that λct≫λtc.

Third, λct>λtc is also required for having a synchronous beating frequency f closer to the *cis* frequency fc [Kamiya and Hasegawa, Exp. Cell. Res. (1987) and Ruffer and Nultsch, Cell Motil. (1985)]. Our minimal model gives f=αfc+(1−α)ft. With fc=45 Hz, ft=65 Hz and f=50 Hz obtained in our own partial deflagellation experiments, λct/λtc≈4. In others’ experiment [see Figure 10 in Wan et al., J. R. Soc. Interface (2013)] fc=57.12 Hz, ft=80.52 Hz and f=63.38 Hz, which also corresponds to λct/λtc≈4. Therefore, this ratio can be determined before even considering the flagellum-to-flow sensitivity.

Altogether, the conditions used in the present modeling, *i.e.*, λct>λtc and both flagella being equally sensitive to external flow drags, capture the experimental phenomena more comprehensively. We have reorganized figures and compared directly the Arnold tongue diagrams of *wt* and *ptx1* in Figure 2B-C:

Line#164: “Albeit noisier, the Arnold tongue of *ptx1* covers a slightly larger width compared to *wt*'s, meaning that the breaststroke beating of two *trans* flagella is similarly entrainable as that by one *cis* and one *trans* (Figure 2B-C right panels). This finding indicates that the *trans* flagellum is at least as susceptible to hydrodynamic loads as the *cis*.”

We also explicitly discuss:

Line#279: “It should be mentioned that our main observation about the measured entrainment strength ε only depending on the *cis*-loading, can also be reproduced by assuming ϵt≈0. This alternative limit implies that the *trans* flagellum has no susceptibility to hydrodynamic loads, which is inconsistent with our entrainment experiments of *ptx1*. In addition, α≈1 is also necessary to explain the difference in noise and effective temperature between the *wt* and *ptx1* experiments.”

B) I find the discussion deriving the Equations S6 and S7 unsatisfactory. Whilst I can agree to consider \δ slaved to the variable \phi0-\phi_f, I do not necessarily agree with the "small \δ" approximation that brings S6 to S7. I agree that the variable \δ can be small, but here the question is whether it is much smaller than (\phi0-\phi_f). This I do not think is necessarily clear a priori. For \δ \sim (\phi0-\phi_f), Equation S7 is not the correct approximation to S6. Notably, the parameters of the resulting equation for d\phi0/dt will depend also e.g. on (\epsilon_c-\epsilon_t). I think the authors should work out this section much more explicitly and justify better their approximations.

We have rewritten Appendix 1 and in particular section Monte-Carlo simulations. We have also simplified our notations and use another set of symbols to denote the phases. In particular, the properties referring to the synchronous flagellar beating (the cell), are now free of subscript; whereas the properties of the external flow now have the subscript “0” (*e.g.*, ϕ0). We have also removed some derivations that were not clear and did not add to the manuscript.

We also realized that there was a typo in Equation S6 (now Equation 7 in Appendix 1), which was missing the term λct−λtc2λΔ˙ on the left hand side. In the updated section, we have simplified the derivation. Linear combinations of Equation S2a,b,c (now Equation 5 in Appendix 1) yield Equations 7 in the new manuscript:

Δ˙=2π[νct−λsin(Δ)−ϵc sin(Ψ+Δ2)+ϵtsin(Ψ−Δ2)]+ζc−ζt,Ψ˙+λct−λtc2λΔ˙=2π[(f−f0)−λctϵcλsin⁡(Ψ+Δ2)−λtcϵtλsin⁡(Ψ−Δ2)]+ζ, with λ=λct+λtc*,*
f=(λctfc+λtcft)λ, ζ=(λctζc+λtcζt)λ and νct=ft−fc. In the asymptotic limit of very large λ (i.e.,λ≫νct,ϵc,t) which represents the case when *cis* and *trans* are very strongly coupled, the solution to the first equation in Equation 7 is Δ˙=0 and Δ=0 (or in other words ϕc=ϕt). In this limit, one directly sees that the second equation above reduces to the Adler equation, from which we obtain ε=(λctϵc+λtcϵt)λ. More details are included in Appendix 1 Sec. Monte-Carlo Simulations – “Interaction between three oscillators”.

C) In the phase oscillator model of Equations S2, the two flagella are assumed to "have the same level of noise (\zeta_c=\zeta_t)" and at the same time their variances show different effective temperatures. I think that this needs to be clarified as the two things do not seem to be compatible to me. (Also, I think there is a typo in Equations S3. The phases on the left hand sides are inverted).

We have corrected the typo in the Equation in Appendix 1 (now Equation 5).

We took ζc=ζt initially to simplify the introduction of the model, but now realize that it only made the description more confusing. We have therefore modified all relevant parts in both Appendix 1 and the main text to reflect our measurements of effective temperature:

Line#464 in Material and Methods: “The noise levels for the *cis*- and the *trans* flagella are taken as Tc, Tt=1.57, 9.42 rad2/s. Under unilateral flagellar coupling, the collective noise level approximates to Tc and the value corresponds to typical experimental observations (31).”

Meanwhile, although there is almost no visual change, the panels in Figure 5 are updated with modeling data generated with the new noise parameters.

D) Throughout the paper the intrinsic frequencies of cis and trans are fixed to f_c=45Hz and f_t=65Hz. However, these parameters will certainly vary from cell to cell. I think it would be good to justify why we can forget about this variability in the analysis. It is not clear to me.

In our own experiments on partially deflagellated cells, we consistently observe flagellar frequencies in quantitative agreement with those reported in the literature (fc, f, ft)=(45Hz , 50Hz , 65Hz), which we used for our study [Kamiya and Hasegawa, Exp. Cell. Res. (1987) and Ruffer and Nultsch, Cell Motil. (1985)]. The variability in f does not affect the analysis and the conclusions of the asymmetric coupling. Our analysis involves scanning the detuning ν and measuring the width of the region where τ>0.5. The location of these regions (absolute value of f) does not affect the measured width (the effective coupling strength ε).

The specific values for fc and ft are used in the present study together with the absolute value of f=(λctfc+λtcft)λ to estimate λct/λtc and we deduce λct/λtc≈4. The important result is that this ratio is significantly larger than one, which characterizes our main conclusion regarding the *cis*-dominance. This conclusion is robust, holding for λct/λtc within a broad range (see figure 5F), and is not affected by variability in fc and ft.

E) If I understand correctly, the load estimates are done with a single waveform, derived from a sample cell. Upon load, the waveforms will change. These (changed) waveforms are the ones that should be used to determine the load. Is this the case in this manuscript? From the Materials and methods this doesn't seem to be the case.

No, the load-computations are based on realistic flagellar shapes, that is, the shapes extracted from long video recording, where the sample cell is subjected to flows along different directions (θ=0∘,45∘,90∘,135∘). There are in total N=4 directions/recordings, each lasting 500 frames (~0.6s, or 30 beats) and for each frame the flagellar shapes are manually extracted to guarantee accuracy. Note that each recording starts with a flash-light event, which marks the onset of the background flow. In this way, the videos also help resolve how the flagellar beating becomes gradually flow-entrained. The present force/power in Figure 1E-G are obtained by averaging over the fully entrained cycles.

To avoid confusion and provide more detail, we have now rewritten the corresponding section in Appendix 1 (now Sec. *Hydrodynamic computation for asymmetric loading*) and provide a figure displaying the dynamic flow-loadings during entrainment.

Also, the readers are reminded:

Line#98: “Our computations provide the loads on the flagella throughout the entire experiment, from the time the captured cell is gradually entrained by the external flow, see Appendix 1: Hydrodynamic computation for asymmetric loading for the entire time series.”

F) The manuscript states that "trans-loads appear to matter negligibly". This needs to be discussed in more detail, given the fact that the ptx1 flagella, which are putatively both trans, can be entrained by the hydrodynamic loads quite well.

We agree. As we have discussed in the reply to the 1^st^ comment, our *ptx1* experiments show that the susceptibility of the *trans* flagellum to external flow is at least as large as that of the *cis*. Our conclusion that *"trans-loads appear to matter negligibly"* is directly related to Equation 3 ε=αϵc+(1−α)ϵt. Even if the *trans* flagella were more entrainable (ϵc<ϵt), the unilateral flagellar coupling (α∼1) still leads to the flow entrainability to be determined by the hydrodynamic loads on the *cis*.

[Editors' note: further revisions were suggested prior to acceptance, as described below.]

The manuscript has been improved but there are some remaining issues that need to be addressed, as outlined below:Reviewer #3 (Recommendations for the authors):I very much appreciate the efforts made by the authors in improving their manuscript. I think the current version is better organised and reads better as well. As I wrote in my previous report, I find their results interesting and valuable and I think this paper should be published.Still, there are a couple of final comments that I would like to put to the authors. I am particularly interested in them addressing point (3) below. The other two are of secondary importance with respect to this one.1) It is still unclear to me how \tau is calculated and what is its relation with the parameters of the Adler model. Isn't there an analytical expression (or at least semi-analytical) that can be put forward? Deciding that \epsilon can be measured from the region \tau\geq0.5 seems a bit arbitrary to me. This boundary seems to work well for the 11 cells tested (see Appendix), but can the method be extrapolated to the general case, e.g. when there is an arbitrary value of the beating noise? I would expect that keeping \epsilon but changing T would decrease "t_entrain" and therefore the width of the region \tau\geq0.5.

We find that using τ>0.5 to measure ε is equivalent to but more robust than previously used methods, when the noise of flagellar beating is low, which is the case for *wt* cells (Quaranta et al., PRL, 2015). In a previous work, we have simultaneously obtained the parameters (T,ε,ν) from the Fokker-Planck Equation, by fitting the probability distribution of Ψ (Polin et al. Science 2009, Quaranta et al., PRL, 2015). However, this multiparameter fitting is not as robust as the simple measure of τ>0.5 used in this study.

Author response image 2 demonstrates how entrainment profiles (obtained with our Monte Carlo simulations) change with increasing noise T, with ϵt,c=2.4Hz. When the noises are low (T≤1.5
rad2/s), where complete entrainment (τ≈1) can be achieved, the measured width determined by τ>0.5 directly reflects the value of ϵt,c, and remains largely invariant against noise. However, as pointed out by the reviewer, under stronger noises (T≥2.5
rad2/s), the use of τ>0.5 becomes ambiguous and tends to underestimate the effective coupling strength ε (*e.g.,* see the entrainment profile for T=6.28 rad2/s).

A semi-analytical expression for the time fraction τ, for which the phase Ψ remains nearly constant, can be approximated by computing the probability that dΨdt remains small. The distribution of dΨdt can be theoretically inferred from the steady state solution Ψ of the associated Fokker-Plank Equation. Here we plot the probability that dΨdt=2π(−ν−εsin⁡Ψ) is smaller than π rad/s, see Author response image 3. We can clearly see that the frequency interval where the probability p(dΨdt≤πrad/s) being larger than 0.5, robustly indicates the value of the forcing ϵt,c, for noise levels smaller than T≤2.5rad2/s. Here again, we see that for larger noise levels this probability no longer reaches 0.5. Accordingly, we have only used this methodology when the noise levels were low.

**Author response image 3. sa2fig3:** 

We now explicitly mention this applicable regime in the manuscript:(Line#149) “This method measures ε accurately when noises are low (T≲2.5rad2/s for typical values of frequencies and couplings used in this work). In this regime, this straightforward method is equivalent to previous methods based on multi-parameter curve fitting (27,31) but is more robust.”

2) Please provide more details on how t_entrain is calculated.

For a given time segment, the flagellar beating is considered entrained if it is phase-locked, i.e., Ψ, the phase difference between the flagella and the flow, varies slow enough over time. Practically, we use time segment of 0.1 s (~5 beats) and we consider the segment entrained if the absolute value of Ψ’s slope is less than π rad/s (equivalent to a frequency mismatch smaller than 0.5 Hz). This threshold is determined to be smaller than our frequency resolution in scanning the detuning (~0.8 Hz).

This information is now added to a new subsection in the *Materials and methods* section:

(Line#464) “Compute time fraction of phase-locking

In practice, phase-locking is considered established if phase difference, either between the two flagella (Δ) or between the flow and flagella (Ψ), varies slow enough over time. Here we use Ψ to illustrate the process. We first break down an entire time series (ttot≈ 10 s) to segments of 0.1 s (~5 beats). A given segment is considered phase-locked if |dΨ / dt| ≤ π rad/s. This particular threshold (π rad/s) is equivalent to a frequency mismatch of 0.5 Hz, which is smaller than our frequency resolution in scanning the detuning (∼ 0.8 Hz).”

3) On pgs. 15,16 the model is compared to the experimental data. Figure 5E shows that starting with \epsilon equal to the experimental value of the frontal flow (2.4Hz) and detuning the frequency of the forcing, f_0, one obtains a synchronisation curve with a width of 2.4Hz. Was this not a given to begin with? I must be missing something. The width of the synchronisation region is precisely given by the value of \epsilon, so I do not understand how this could have given anything different. The same goes for Figure 5C,D, where it is clear that the boundaries of the Arnold's tongue are straight lines at 45 degrees.The experiments have been analysed in terms of an Adler model. The parameters are then inserted in a model whose effective dynamics is equivalent to that of an Adler model. We should then expect to recover the behaviour of the experiments. This seems to me to be a bit of a circular argument and I do not understand how the results of Figure 5 could have been any different. I think it would be very helpful if the authors could clarify this point in the paper.

Just like the reviewer has pointed out, for the entrainment profile of frontal flow (the yellow line in the lower panel of Figure 5E), the measured half width (ε), 2.4 Hz, directly corresponds to the model setting ϵc=ϵt= 2.4 Hz. This simulation confirms that the experimental entrainment profile of frontal-flow can be reproduced, and more importantly, it works as a reference for the profiles of *cis-* and *trans-*flows in the lower panel of Figure 5E.

In this panel showing modeling results, we primarily want to show: selectively loading the leading flagellum (*cis*) broadens the entrainment profile (red data) while selectively loading the follower (*trans*) narrows it (blue data). These broadening and narrowing effects capture the experimental phenomenology and their only prerequisite is a *cis*-dominated asymmetric coupling. This helps us arrive at the conclusion of this manuscript.

Lastly, in Figure 5CD, our primary goal of reproducing Arnold tongues is to highlight the effect of stronger noise in the *ptx1*, *i.e.*, it makes the edges of the tongue less well-defined. This shows that our model is able to reproduce the different experiments with different mutants.

The following changes are made in the text to address the ambiguity pointed out by the Reviewer:

(Line#310) (for Figure 5E) “[…with the selective forcing, the model reproduces experimental observations that] the entrainment profile of *cis*-flow is consistently broader than that of frontal-flow (i.e., larger ε), and the profile of *trans*-flow is always the narrowest (smallest ε).”

(Line#301) (for Figure 5CD) “Compared with *ptx1*, the asymmetry λct≫λtc does not affect the overall shape of the Arnold tongue but leads to a much lower noise level in the case of wt, which is essentially *cis*-induced.”

We have also re-emphasized the difference between the measured coupling strength ε and the model parameters ϵc,t:

(Line#136) “In this study, ε is experimentally measured from the phase dynamics to quantify the flow entrainment effectiveness.”

(Line#263) “In this simple model, we can differentially vary ϵc and ϵt to match the values of the selective hydrodynamic loads (F¯Flowc/F¯Flowt) measured for each flow condition (Figure 1E-G). It bears emphasis that ϵc,t are input parameters of our model, whereas ε is measured from the phase dynamics to characterize the entrainment. In our simulations, ε is extracted following the same approach as in the experiments, see Figure 2A.”